# The Matsuno baroclinic wave test case

Ofer Shamir[1], Itamar Yacoby[1], Shlomi Ziskin Ziv[1], and Nathan Paldor[1]

[1]Fredy and Nadine Herrmann Institute of Earth Sciences, Edmond J. Safra Campus, Givat Ram, The Hebrew University of Jerusalem, Jerusalem, Israel

**Correspondence:** Nathan Paldor (nathan.paldor@huji.ac.il)

**Abstract.** The analytic wave-solutions obtained by Matsuno (1966) in his seminal work on equatorial waves provide a simple and informative way of assessing the performance of atmospheric models by measuring the accuracy with which they simulate these waves. These solutions approximate the solutions of the shallow water equations on the sphere for small speeds of gravity waves such as those of the baroclinic modes in the atmosphere. This is in contrast to the solutions of the non-divergent barotropic vorticity equation, used in the Rossby-Haurwitz test case, which are only accurate for large speeds of gravity waves such as those of the barotropic mode. The proposed test case assigns specific values to the wave-parameters (gravity wave speed, zonal wave-number, meridional wave-mode and wave-amplitude) for both planetary and inertia gravity waves, and suggests simple assessment criteria suitable for zonally propagating wave solutions. The test is successfully applied to a spherical shallow water model in an equatorial channel and to a global-scale model. By adding a small perturbation to the initial fields it is demonstrated that the chosen initial waves remain stable for at least 100 wave-periods. The proposed test case can also be used as a resolution convergence test.

## 1 Introduction

A cornerstone of global-scale model assessment is the Rossby-Haurwitz test case, originally used by Phillips (1959) as a qualitative way of assessing his shallow water model. Phillips initialized his model with an analytic wave-solution of the non-divergent barotropic vorticity equation obtained by Haurwitz (1940), and examined the spatio-temporal smoothness of the simulated fields at later times. Using this procedure he concluded that the emergent noise in his model was due to a small-but-significant, divergence field missing from the initial fields. Even though the solutions of the non-divergent barotropic vorticity equation are not solution of the Shallow Water Equations (SWEs), Phillips' procedure was adopted by Williamson et al. (1992) as a standard test case for shallow water models and has been extensively used ever since (Jablonowski et al., 2009; Mohammadian and Marshall, 2010; Bosler et al., 2014; Ullrich, 2014; Li et al., 2015, are only five recent examples).

However, there are two known issues with the original Rossby-Haurwitz test case that limit its usefulness (Thuburn and Li, 2000). The first is the generation of small-scale features via potential enstrophy cascade, which requires adequate dissipation mechanisms to remove enstrophy at the grid scale (in order to mimic a continuous cascade to sub-grid scales). The second is

the instability of the initial wave-number 4 used in the Rossby-Haurwitz test case. In contrast to Hoskins (1973) who found that wave-numbers smaller than or equal to 5 are stable, Thuburn and Li show that Rossby-Haurwitz wave-number 4 is in fact also unstable.

Recently, Shamir and Paldor (2016) proposed a similar procedure to that of Phillips (1959) where instead of using the solutions of the non-divergent barotropic vorticity equation, the initial fields are the analytic wave-solutions of the linearized SWEs on the sphere derived in Paldor et al. (2013). These solutions fully account for the small divergence field and can be computed on any grid given the locations of the latitudes and longitudes. In particular, they include the fast propagating Inertia- Gravity (IG) waves that are completely absent from the the non-divergent barotropic vorticity equation. Consequently, the procedure proposed by Shamir and Paldor (2016) provides a more quantitative assessment than Phillips's original procedure while it is just as easy to implement.

Both solutions obtained by Haurwitz (1940) and Paldor et al. (2013) approximate the solutions of the SWEs in the asymptotic limit of large speed of gravity waves. For most practical purposes they are sufficiently accurate for speeds of gravity waves of about $200 - 300$ ms$^{-1}$ or higher, which are typical of the barotropic mode in Earth's atmosphere and oceans. However, typical speeds of gravity waves of baroclinic modes in the (tropical) atmosphere are about $20 - 30$ ms$^{-1}$ (Wheeler and Kiladis, 1999). Thus, the above procedures are only relevant for assessing the accuracy with which the barotropic wave mode is simulated. In order to assess the accuracy of the baroclinic wave modes we propose, in the present work, to use the analytic wave-solutions of the linearized SWEs on the equatorial $\beta$-plane obtained by Matsuno (1966) that approximate the solutions of the SWEs on the sphere in the asymptotic limit of small speed of gravity waves (De-Leon and Paldor, 2011; Garfinkel et al., 2017).

In addition to being on two opposite ends of the spectrum of gravity wave speed the solutions obtained by Matsuno (1966) differ from those obtained by both Haurwitz (1940) and Paldor et al. (2013) in their meridional extent. While the former become negligibly small outside a narrow equatorial band the latter two have non-negligible amplitudes in the vicinity of the poles. Thus, while the Rossby-Haurwitz test case is only relevant to global-scale models, the test case proposed in the present study is applicable to both global-scale and tropical models.

A homonymous, but unrelated, test case is the baroclinic wave test case developed in Jablonowski (2004) and Jablonowski and Williamson (2006) and independently in Polvani et al. (2004), and its variants in Lauritzen et al. (2010) and Ullrich et al. (2014). This test case is concerned with the non-linear generation of synoptic-scale eddies in multi-layer models via baroclinic instability. In contrast, the test case proposed here is concerned with linear wave propagation in (non-linear) single-layer models. In particular, while the term baroclinic usually implies the use of multi-layer models, here this term is used to denote a single thin layer model of homogeneous density where the gravity waves speeds are similar to those observed in baroclinic modes in the atmosphere.

The idea of using Matsuno's solutions as a test case in a similar fashion to that of the Rossby-Haurwitz test case is most likely not original, but has never been standardized. Thus, the purpose of the present work is to standardize the Matsuno test case in the same spirit that Williamson et al. (1992) standardized the Rossby-Haurwitz one. We start with a short description of the analytic expressions derived by Matsuno (1966) in section 2. The proposed test procedure, including the choice of wave-parameters and assessment criteria, is described in Section 3. In section 4 we demonstrate the usefulness of the proposed

test case using both an equatorial channel spherical shallow water model, and a global-scale one. In addition, we examine the smoothness and stability of the initial waves in a similar fashion to that used in Thuburn and Li (2000) and demonstrate the possibility of using the proposed test case as a resolution convergence test. The paper ends with some concluding remarks in section 5.

## 2   The analytic solutions

The proposed test case is based on the analytic solutions of the SWEs on the equatorial $\beta$-plane obtained by Matsuno (1966). These solutions have the form of zonally propagating waves, i.e.

$$
\begin{bmatrix} u(x,y,t) \\ v(x,y,t) \\ \Phi(x,y,t) \end{bmatrix} = \mathrm{Re} \left\{ \begin{bmatrix} \hat{u}(y) \\ \hat{v}(y) \\ \hat{\Phi}(y) \end{bmatrix} e^{i(kx-\omega t)} \right\} \tag{1}
$$

where $x$ and $y$ are the local Cartesian coordinates in the zonal and meridional directions, respectively; $t$ is time; $u$ and $v$ are the velocity components in the zonal and meridional directions, respectively; $\Phi$ is the geopotential height; $k$ is the planar zonal wave-number (which has dimensions of m$^{-1}$); $\omega$ is the wave-frequency; $\hat{u}(y), \hat{v}(y)$ and $\hat{\Phi}(y)$ are the latitude dependent amplitudes; and $i = \sqrt{-1}$ is the imaginary unit. In accordance with the sign convention used in Matsuno we assume $k$ is non-negative and let $\omega$ take any real value. Note, however, that the sign in front of $\omega$ in (1) is opposite to that in Matsuno's theory. The convention chosen here is more intuitive as it implies that positive values of $\omega$ correspond to waves that propagate in the positive $x$ direction, i.e. eastward.

The unknown wave-frequencies and latitude dependent amplitudes are derived from the (well-known) energies and eigenfunctions of the (time-independent) Schödinger equation of quantum harmonic oscillator. The resulting frequencies are given by the solutions of the following cubic equation

$$
\omega_{n,k}^3 - \left[ gHk^2 + \frac{2\Omega\sqrt{gH}}{a}(2n+1) \right] \omega_{n,k} - \frac{2\Omega gHk}{a} = 0, \tag{2}
$$

for $n = -1, 0, 1, 2, \ldots$, where $\Omega = 7.29212 \cdot 10^{-5}$ rad s$^{-1}$, $a = 6.37122 \cdot 10^6$ m and $g = 9.80616$ m s$^{-2}$ are the Earth's angular frequency, mean radius, and gravitational acceleration respectively; and $H$ is the mean layer's depth (thickness).

For $n \geq 1$ Equation (2) has three distinct real roots corresponding to a slowly westward propagating Rossby wave, a fast Eastward propagating Inertia Gravity (EIG) wave, and a fast Westward propagating Inertia Gravity (WIG) wave. For $n = 0$ one of the three roots, the one corresponding to a westward propagating gravity wave with $\omega = -\sqrt{gH}k$, leads to infinite zonal wind and is thus discarded as a physically reasonable solution. The remaining two roots correspond to the lowest (i.e. $n = 0$) EIG wave and the Mixed Rossby-Gravity (MRG) wave. For $n = -1$ Equation (2) has one real root $\omega = \sqrt{gH}k$, which correspond to the equatorial Kelvin wave (see Matsuno, 1966). The existence of the latter two waves on a sphere is discussed in Garfinkel et al. (2017) and Paldor et al. (2018)

For given values of the zonal wave-number, $k$, and meridional mode-number, $n$, the roots of the cubic equation can be obtained in a closed analytic form using the solutions of the general cubic equation as follows (e.g. Abramowitz and Stegun,

1964):

$$\omega_{n,k,j} = \text{Re}\left\{ -\frac{1}{3}\left(\Delta_j + \frac{\Delta_0}{\Delta_j}\right)\right\}, \quad \text{for} \quad j = 1,2,3 \tag{3}$$

where $j$ stands for the three roots, and where

$$\Delta_0 = 3\left[gHk^2 + \frac{2\Omega\sqrt{gH}}{a}(2n+1)\right], \tag{4a}$$

$$\Delta_j = \left[\frac{\Delta_4 + \sqrt{\Delta_4^2 - 4\Delta_0^3}}{2}\right]^{1/3} \exp\left(\frac{2\pi j}{3}i\right), \tag{4b}$$

$$\Delta_4 = -\frac{54\Omega gHk}{a}. \tag{4c}$$

Given the definitions in (4), the explicit expressions for the frequencies of the Rossby, WIG and EIG waves are obtained by sorting the values in (3) as follows:

Rossby : $$\omega_{n,k,\text{R}} = -\min_{j=1,2,3}|\omega_{n,k,j}|, \tag{5a}$$

Westward Inertia-Gravity : $$\omega_{n,k,\text{WIG}} = \min_{j=1,2,3}\omega_{n,k,j}, \tag{5b}$$

Eastward Inertia-Gravity : $$\omega_{n,k,\text{EIG}} = \max_{j=1,2,3}\omega_{n,k,j}. \tag{5c}$$

Having found (one of) the wave-frequencies for a given combination of $n$ and $k$, the corresponding latitude dependent amplitudes can be written as

$$\hat{v}_n = A\hat{H}_n\left(\epsilon^{1/4}\frac{y}{a}\right)\exp\left(-\frac{1}{2}\left(\epsilon^{1/4}\frac{y}{a}\right)^2\right) \tag{6a}$$

$$\hat{u}_{n,k} = \frac{gH\epsilon^{1/4}}{ia(\omega_{n,k}^2 - gHk^2)}\left[-\sqrt{\frac{n+1}{2}}\left(\frac{\omega_{n,k}}{\sqrt{gH}} + k\right)\hat{v}_{n+1} - \sqrt{\frac{n}{2}}\left(\frac{\omega_{n,k}}{\sqrt{gH}} - k\right)\hat{v}_{n-1}\right] \tag{6b}$$

$$\hat{\Phi}_{n,k} = \frac{gH\epsilon^{1/4}}{ia(\omega_{n,k}^2 - gHk^2)}\left[-\sqrt{\frac{n+1}{2}}\left(\omega_{n,k} + \sqrt{gH}k\right)\hat{v}_{n+1} + \sqrt{\frac{n}{2}}\left(\omega_{n,k} - \sqrt{gH}k\right)\hat{v}_{n-1}\right], \tag{6c}$$

for $n = 1,2,3,\ldots$ (the cases $n = -1,0$ require special treatment), where $\epsilon = (2\Omega a)^2/gH$ is Lamb's parameter, $A$ is an arbitrary amplitude (that has dimensions of m s$^{-1}$), and $\hat{H}_n$ are the normalized Hermite polynomials of degree $n$ defined by the following three-term recurrence relation (Press et al., 2007)

$$\hat{H}_{-1}(x) = 0, \tag{7a}$$

$$\hat{H}_0(x) = \pi^{-1/4}, \tag{7b}$$

$$\hat{H}_{n+1}(x) = x\sqrt{\frac{2}{n+1}}\hat{H}_n - \sqrt{\frac{n}{n+1}}\hat{H}_{n-1}. \tag{7c}$$

Note: (i) The chosen normalization for the latitude dependent amplitudes in (6) is different from the one used in Matsuno. We use the above normalization for convenience, as it guarantees that $\hat{v}$ is independent of both $k$ or $\omega$. (ii) The use of the normalized version of the Hermite polynomials also leads to slightly different pre-factors in front of $\hat{v}_{n+1}$ and $\hat{v}_{n-1}$ compared to Matsuno. However, they are generally more computationally stable. (iii) The outer parentheses in (6a) denote the argument of $\hat{H}_n$ and the exponential function, and not multiplicative factors. In other words, the independent variable in this equation is $(\epsilon^{1/4}y/a)$, and not simply $y$.

While the solutions obtained by Matsuno (1966) apply for the equatorial $\beta$-plane, the proposed test case is intended for use in spherical models. As is shown in Garfinkel et al. (2017), the SWEs on the equatorial $\beta$-plane approximate the SWEs on the sphere to zero-order in powers of $1/\epsilon^{1/4}$. Thus, the solutions obtained by Matsuno are only accurate in the asymptotic limit $\epsilon \to \infty$. For the fixed values of Earth's angular frequency and mean radius, this implies that the solutions obtained by Matsuno are only accurate for sufficiently small speeds of gravity waves $\sqrt{gH}$.

In practice, in order to use Matsuno's solutions in spherical models, the local Cartesian coordinates $x$ and $y$ in the above formulae (1) and (6) have to be replaced by the longitude $\lambda$ and latitude $\phi$ of the geographical coordinate system. Recall that the transformation from the Cartesian system to the spherical one is $(x, y) \to a(\cos\phi_0\lambda, \phi)$, where $\phi_0$ is the central latitude at which the planar approximation is applied. Likewise, the planar wave-number $k$ in formulae (1)-(6) has to be replaced by its spherical counterpart, $k_s$, using the transformation $k \to k_s/a\cos\phi_0$. Thus, for the equatorial $\beta$-plane where $\phi_0 = 0$, the transformation is simply $(x, y) \to a(\lambda, \phi)$ and $k \to k_s/a$. In particular, the reader should note that the planar wave-number $k$ has units of m$^{-1}$ while the spherical wave-number $k_s$ is dimensionless.

## 3 The proposed test procedure

The general procedure of the proposed test case is similar to the Rossby-Haurwitz one in that the model in question is initialized with velocity and height fields corresponding to a particular wave-solution and the time evolution of that wave is then examined. The initial wave fields in this case are taken from the analytic expressions in Section 2. The specific choice of wave-parameters and assessment criteria in the present work are discussed below, separately. As is often the case, these choices represent compromises between conflicting factors, e.g. adherence to observations vs. adherence to asymptotic validity of the analytic solutions or rigorous testing vs. simplicity. In any case, these choices may be the subject of discourse as deemed appropriate by the community.

### 3.1 wave-parameters

The wave-parameters consist of the speed of gravity waves, $\sqrt{gH}$, the wave-number and wave-mode, $k$ and $n$, the wave-amplitude, $A$, and the wave-type. Any given combination of these parameters completely specifies a unique wave using the expressions in (1)-(6). We consider all other parameters, including the spatio-temporal resolution and the form of diffusion/viscosity terms, to be modeling choices left to the developers. This approach is aimed at testing the models in their modus

operandi. However, as noted in Polvani et al. (2004), different choices of the form of diffusion/viscosity terms correspond to different sets of equations and may not converge to the same solutions.

The choice of gravity wave speed $\sqrt{gH}$ is inspired by the observed speed of gravity waves of the baroclinic modes in the atmosphere. In practice, we keep $g$ fixed to Earth's gravitational acceleration and set the speed of gravity waves by letting $H = 30$ m, which is within the range of observed equivalent depths in the equatorial atmosphere (Wheeler and Kiladis, 1999). As mentioned in section 2, the analytic solutions obtained by Matsuno on the equatorial $\beta$-plane are only accurate approximations of the SWEs on the sphere in the asymptotic limit of small speeds of gravity waves. The above value was found by trial and error to be sufficiently accurate in the sense that it yields stable solutions for at least 100 wave-periods in the simulations described in Section 4.

In addition to the speed of gravity waves, the accuracy of Matsuno's solutions depends also on the wave-number and wave-mode. For a given value of $\sqrt{gH}$, these solutions become asymptotically accurate in the limits $k_s, n \to 0$ (but $k_s \neq 0$) (De-Leon and Paldor, 2011). Also, the spatial variability and the required spatial resolution both increase with the wave-number or wave-mode so both of these considerations suggest that reasonable choices for the wave-number and wave-mode consist of small to moderate values. The proposed wave-number and wave-mode are $k_s = 5$ (i.e. $k = 5/a$ m$^{-1}$) and $n = 1$, i.e. within the range of dominant values observed in the equatorial atmosphere (Wheeler and Kiladis, 1999), but other choices may work just as well provided $k_s$ and $n$ are not too large.

The proposed test case is based on the solutions of the linear SWEs but is intended to be used in non-linear models. Therefore, the waves-amplitude should be sufficiently small so as to satisfy the linearization condition. The proposed amplitude of $\hat{v}$ in Equation (6) is $A = 10^{-5}$ m s$^{-1}$, chosen by trial and error so as to enable stable solutions for at least 100 wave periods in the simulations of Section 4.

In general, there are two qualitatively different wave types, Rossby and IG, that differ in the magnitude of their divergence and vorticity fields. The former is more solenoidal (non-divergent), whereas the latter is more irrotational. In order to assess the models' performances in these two qualitatively different limits we suggest using one of each. Since Rossby waves are exclusively westward propagating, we choose the EIG wave of the two IG waves as the second case to cover the two directions of longitudinal propagation.

For these chosen values of $\sqrt{gH}$, $k$ and $n$ the wave-periods, $T = \frac{2\pi}{\omega}$, are $T = 18.5$ days for the Rossby wave and $T = 1.9$ days for the EIG wave.

## 3.2 Assessment criteria

For sufficiently small wave-amplitudes we expect the spatio-temporal structure of the simulated solutions to be that of zonally propagating waves, i.e. $q = \hat{q}(\phi)e^{i(k\lambda - \omega t)}$ (where $q$ stands for any of the dependent variables $u$, $v$ or $\Phi$), with frequency and latitude dependent amplitudes corresponding to the initial wave. In this case, it is desirable to assess the accuracy of the zonal and meridional structures of the waves independently. A fast and simple way of doing so is using Hovmöller diagrams, where the temporal change in any direction is isolated by intersecting the fields along a fixed value of the other direction. This results in the following two diagrams:

(i) A time-longitude diagram obtained by intersecting the fields at a certain latitude. The contour lines in the time-longitude plane are the set of points satisfying $k\lambda - \omega t = const$ (for some real $const$). Thus, the expected pattern for this diagram is that of straight lines with slopes that equal the inverse of the wave's phase speed $k/\omega$. In order to avoid small fluctuations in the vicinity of latitudinal zero-crossings, we recommend using latitudinal intersects at or near local extrema.

(ii) A latitude-time diagram obtained by intersecting the fields at a certain longitude. For any two wave-fronts with an equal phase $k(\lambda_2 - \lambda_1) = \omega(t_2 - t_1)$. Thus, holding $\lambda$ fixed while varying $t$ from $t_1$ to $t_2$ is equivalent to holding $t$ fixed and varying $\lambda$ from $\lambda_1$ to $\lambda_2 = \lambda_1 + \omega/k(t_2 - t_1)$. The resulting pattern is similar to that of a latitude-longitude diagram, but provides an estimate of the time evolution at a particular longitude (as opposed to a latitude-longitude snapshot at a particular time).

Likewise, for zonally propagating waves it is also desirable to isolate the errors in the phase speed and spatial structure. As
discussed in Shamir and Paldor (2016), the frequently used spherical $l_2$ error entangles the two, and is therefore of lesser use for assessing the accuracy with which the model simulates a propagating wave. Thus for a more quantitative assessment we suggest using the relative difference between the Root-Mean-Square of the analytic solution and the simulated solutions, i.e.

$$\frac{\sqrt{I[q^2]} - \sqrt{I[q_a^2]}}{\sqrt{I[q_a^2]}}, \tag{8}$$

where the quantities $q$ and $q_a$ (which can be vectors) correspond to the simulated and analytic solutions, respectively, and where

$$I[q] = \frac{1}{4\pi} \int\limits_{0}^{2\pi} \int\limits_{-\pi/2}^{\pi/2} q(\lambda, \phi) \cos\phi \, d\phi \, d\lambda. \tag{9}$$

Henceforth we refer to the quantity in (8) as the structure-error since, as opposed to the $l_2$ error, it is unaffected by phase speed errors (i.e. phase shifts in $\lambda$).

## 4   Results

In this section we demonstrate the usefulness of the Matsuno test case by applying the proposed procedure to both an equatorial channel finite-difference model and a global-scale spectral one. We then examine the stability of the selected waves/modes in a similar fashion to that used in Thuburn and Li (2000) for the wave-number 4 Rossby-Haurwitz wave. Finally, we demonstrate the possibility of using the analytic solutions obtained by Matsuno as a resolution convergence test.

## 4.1 Demonstration using an equatorial channel finite-difference model

The model is a spherical version of the Cartesian model used in Gildor et al. (2016), in which the integration forward in time is carried out using the conservation form of the SWEs

$$\frac{\partial U}{\partial t} + \frac{1}{a\cos\phi}\frac{\partial}{\partial\lambda}\left(\frac{U^2}{h}\right) + \frac{1}{a}\frac{\partial}{\partial\phi}\left(\frac{UV}{h}\right) - \frac{2UV\tan\phi}{ah} - 2\Omega\sin\phi V = -\frac{g}{2a\cos\phi}\frac{\partial h^2}{\partial\lambda} \tag{10a}$$

$$\frac{\partial V}{\partial t} + \frac{1}{a\cos\phi}\frac{\partial}{\partial\lambda}\left(\frac{UV}{h}\right) + \frac{1}{a}\frac{\partial}{\partial\phi}\left(\frac{V^2}{h}\right) - \frac{(U^2 - V^2)\tan\phi}{ah} + 2\Omega\sin\phi U = -\frac{g}{2a}\frac{\partial h^2}{\partial\phi} \tag{10b}$$

$$\frac{\partial h}{\partial t} + \frac{1}{a\cos\phi}\left[\frac{\partial U}{\partial\lambda} + \frac{\partial(V\cos\phi)}{\partial\phi}\right] = 0, \tag{10c}$$

where $U = hu$, $V = hv$ and $h$ is the total layer thickness. The numerical scheme employs a standard finite difference shallow-water solver in which the time-differencing follows a leapfrog scheme (centre differencing in both time and space). The computations were done on an Arakawa C-grid. The model contains provisions for a temporal Robert-Asselin filter, but the filter's coefficient was set to zero in the simulations of the present section. In addition, the model includes no diffusion/viscosity terms.

The computational domain is $-180° \leq \lambda \leq 180°$ and $-30° \leq \phi \leq 30°$. The boundary conditions are periodicity at the zonal boundaries $\lambda = \pm 180°$ and vanishing meridional velocity at the channel's boundaries $\phi = \pm 30°$. The corresponding values of $h$ and $U$ at the boundaries are determined by the differential equations. For the chosen wave-parameters the amplitude of the meridional velocity $\hat{v}$ in (6) has an e-folding latitude of $11°$, and its amplitude at $\phi = \pm 30°$ decays to $4e-03$ of its maximal value, so the velocity outside the computational domain can be comfortably neglected. The grid-spacing and time step are $\Delta\lambda = \Delta\phi = 0.5°$ and $\Delta t = 600$ seconds, which were found to yield stable solutions for at least 100 wave-periods.

Figure 1 shows the initial (top row) $u, v, \Phi, \xi, \delta$ fields (where $\xi$ and $\delta$ are the relative vorticity and divergence, respectively) of the chosen Rossby wave mode, and the resulting latitude-time (middle row) and time-longitude (bottom row) Hovmöller diagrams of the simulated solution. The initial fields were obtained using the analytic expressions of Section 2 and wave-parameters of Section 3.1. The simulated solutions were obtained using the above equatorial channel model. The chosen intersects used in the calculation of the Hovmöller diagrams are indicated by white dashed lines superimposed on the initial fields, and are also provided in the Figure's caption. For the sake of legibility the shown time domain in each panel is only the last wave periods of the simulation, i.e. $99T \leq t \leq 100T$, where $T$ is the wave-period. The fields are normalized on their global maximum at $t = 0$. Thus, white regions correspond to times at which the simulated solution exceeds the initial wave-amplitude, momentarily. With this in mind, recall that the patterns in the latitude-time diagrams are similar to those of a latitude-longitude diagram, and can therefore be used to compare with the initial fields. In general, the initial wave-structure is preserved and the dominant slope in the time-longitude diagrams corresponds to the analytic slope indicated with dashed white lines (bottom row). There are, however, some noticeable deviations: A slight east-west tilt can be observed in the latitude-time diagrams (middle row), but most egregiously, the divergence field is less regular than the other four. We return to this last point at the

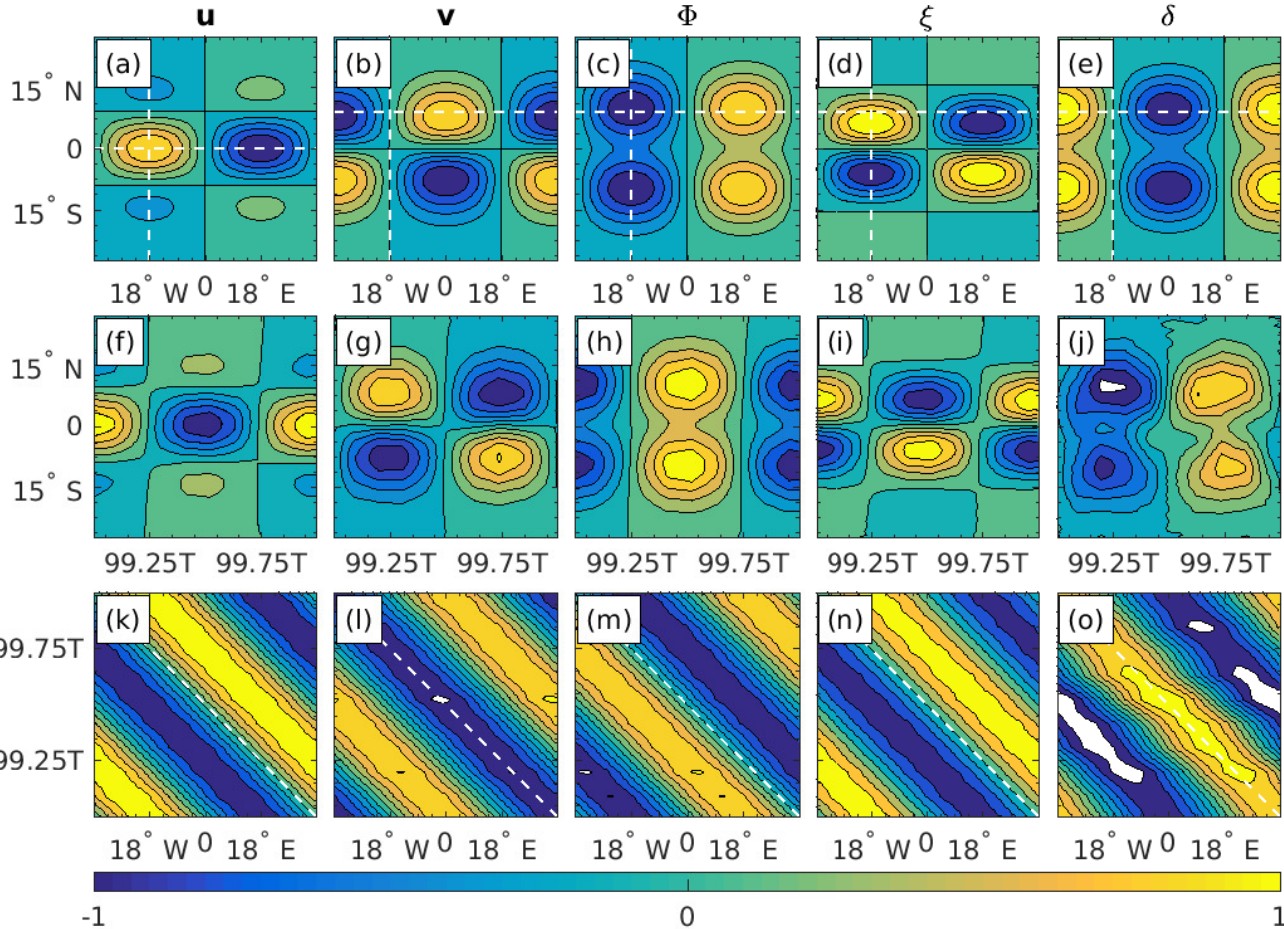

**Figure 1.** Top row: the initial $u, v, \Phi, \xi, \delta$ Rossby wave fields (top row), obtained using the analytic expressions of Section 2 and wave-parameters of Section 3.1. Middle row: latitude-time Hovmöller diagrams of the simulated solutions, obtained by intersecting the fields at $\lambda = -18°$ (also indicated by white vertical dashes lines in the top row). Bottom row: time-longitude Hovmöller diagrams of the simulated solutions, obtained by intersecting $u$ at $\phi = 0°$ and all other fields at $\phi = 9°$ (also indicated by white horizontal dashes lines in the top row). The simulated solutions were obtained using the equatorial channel finite difference model. The fields are normalized on their global maximum at $t = 0$. The wave-period for the chosen wave-parameters is $T = 18.5$ days. Contour-levels range from $-1.0$ to $+1.0$ by $0.2$.

end of Section 4.3. The phase of the simulated patterns in the latitude-time diagrams fit the expected patterns considering the westward propagation of the Rossby mode at $\lambda = -18°$ in one wave-period after 99 wave-periods.

Similarly, Figure 2 shows the initial (top row) $u, v, \Phi, \xi, \delta$ fields of the chosen EIG wave mode, and the resulting latitude-time (middle row) and time-longitude (bottom row) Hovmöller diagrams of the simulated solution. Note that under the normalization used here the initial $v$ field is independent of the wave type and is therefore identical to the one in Figure 1. In contrast to Figure 1 the patterns in the latitude-time diagrams of the simulated solutions are noticeably out of phase. However, considering the

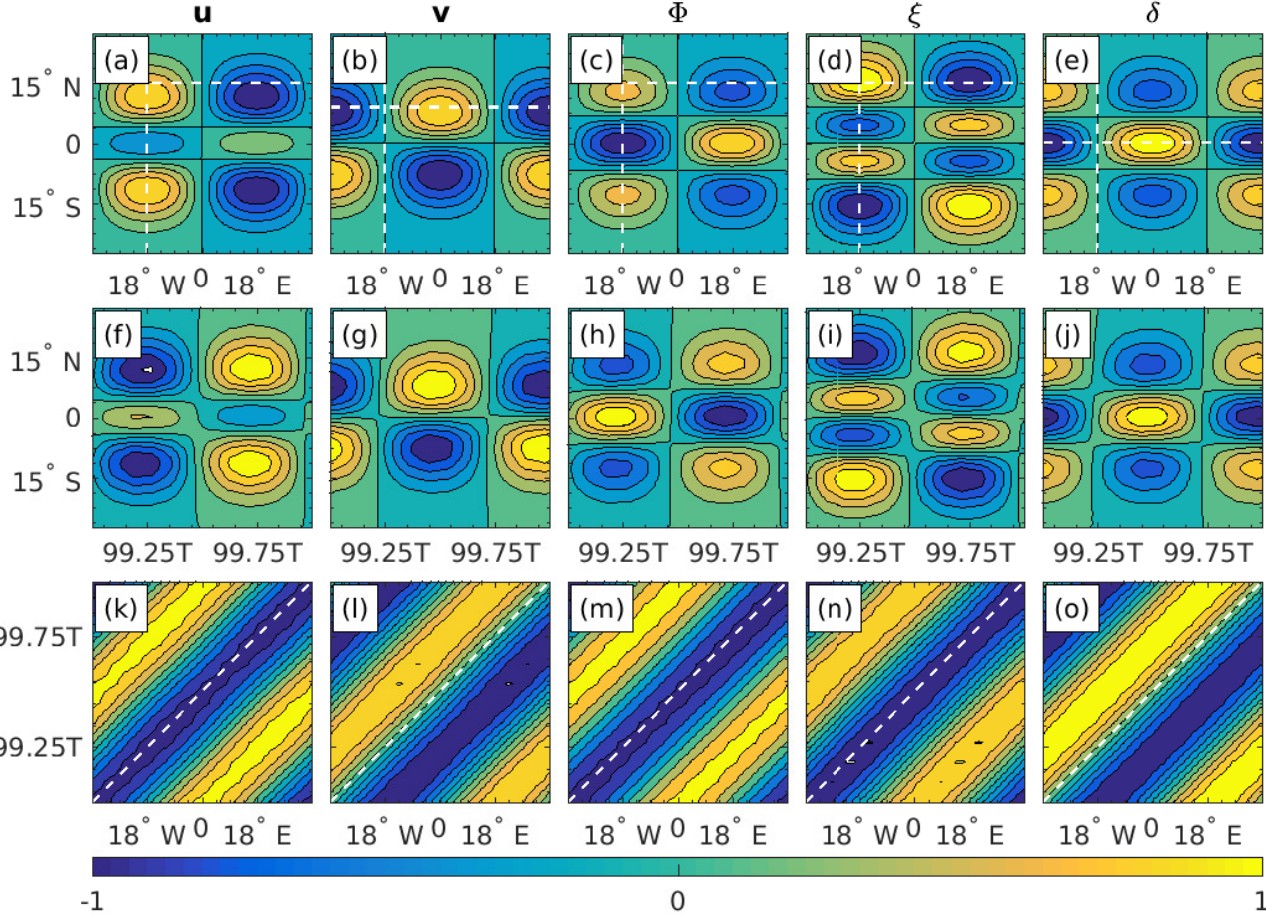

**Figure 2.** Top row: the initial $u, v, \Phi, \xi, \delta$ EIG wave fields (top row), obtained using the analytic expressions of Section 2 and wave-parameters of Section 3.1. Middle row: latitude-time Hovmöller diagrams of the simulated solutions, obtained by intersecting the fields at $\lambda = -18°$ (also indicated by white vertical dashes lines in the top row). Bottom row: time-longitude Hovmöller diagrams of the simulated solutions, obtained by intersecting $v$ at $\phi = 9°$, $\delta$ at $\phi = 0°$ and all other fields at $\phi = 15°$ (also indicated by white horizontal dashes lines in the top row). The simulated solutions were obtained using the equatorial channel finite difference model. The fields are normalized on their global maximum at $t = 0$. The wave-period for the chosen wave-parameters is $T = 1.9$ days. Contour-levels range from $-1.0$ to $+1.0$ by $0.2$.

agreement between the dominant slope in the time-longitude diagrams and the analytic slope indicated by the dashed white lines (bottom row), it is reasonable to say that this phase shift results only from a small phase speed error that accumulates over time. In addition, in contrast to the Rossby wave in Figure 1, the divergence field in this case is just as regular as the other four fields.

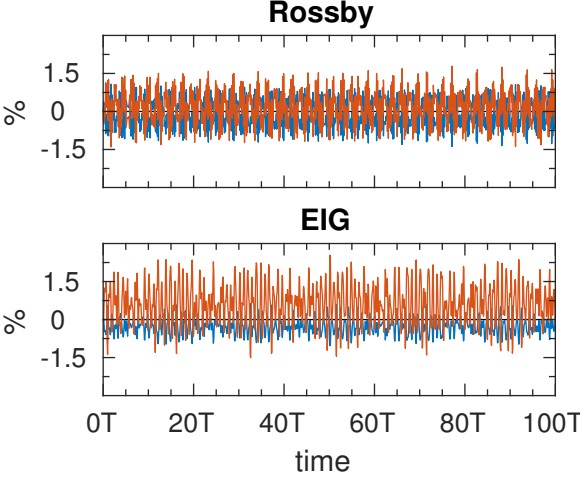

**Figure 3.** The structure-error defined in (8) for both the Rossby (top) and EIG (bottom) waves as a function of time. Blue: calculated for the velocity vector $\mathbf{u} = (u, v)$. Red: calculated for the geopotential $\Phi$.

The structure-error defined in (8) is shown in Figure 3 for both Rossby (top) and EIG (bottom) waves as a function of time. In both cases the structure-error fluctuates about a mean value of less than $1\%$ and there is no visible trend throughout the simulation time of 100 wave-periods. Recall that the structure-error defined in (8) is insensitive to phase differences.

### 4.2 Demonstration using a global-scale spectral model

To demonstrate the applicability of the Matsuno wave as a test case for global-scale models we use the Geophysical Fluid Dynamics Laboratory's (GFDL's) spectral transformed shallow water model which uses the Spherical Harmonics as its basis functions (https://www.gfdl.noaa.gov/idealized-spectral-models-quickstart/). The chosen spectral resolution was T85, i.e. a triangular truncation where both the highest retained wave-number and the total wave-number equal 85. The chosen time step was $\Delta t = 600$ seconds, as in the equatorial channel model. The model contains provisions for hyper-diffusion terms as well as a temporal Robert-Asselin filter, but the coefficients of both were set to zero for the simulations described below.

Figure 4 shows the initial (top row) $u, v, \Phi, \xi, \delta$ fields (where $\xi$ and $\delta$ are the relative vorticity and divergence, respectively) of the chosen Rossby wave mode, and the resulting latitude-time (middle row) and time-longitude (bottom row) Hovmöller diagrams of the simulated solution. The initial fields were obtained using the analytic expressions of Section 2 and wave-parameters of Section 3.1. The simulated solutions were obtained using the above GFDL's global-scale spectral model. The chosen intersects used in the calculation of the Hovmöller diagrams are indicated by white dashed lines superimposed on the initial fields, and are also provided in the Figure's caption. For the sake of legibility the shown time domain in each panel is only the last wave period of the simulation, i.e. $99T \leq t \leq 100T$, where $T$ is the wave-period. The fields are normalized on their global maximum at $t = 0$. Thus, white regions correspond to times at which the simulated solution exceeds the initial wave-amplitude, momentarily. With this in mind, recall that the patterns in the latitude-time diagrams are similar to those of a

latitude-longitude diagram, and can therefore be used to compare with the initial fields. Indeed, the patterns in the latitude-time diagrams of the simulated solutions agree quite accurately with those of the initial wave-structure, but are noticeably out of phase. Nevertheless, considering the agreement between the dominant slope in the time-longitude diagrams and the analytic slope indicated with dashed white lines (bottom row), it is reasonable to say that this phase shift results from a small phase

speed error that accumulates over time. In addition, the divergence field is less regular than the other four fields. We return to this point at the end of Section 4.3.

Similarly, Figure 5 shows the initial (top row) $u, v, \Phi, \xi, \delta$ fields of the chosen EIG wave mode, and the resulting latitude-time (middle row) and time-longitude (bottom row) Hovmöller diagrams of the simulated solution. Note that under the normalization used in the present paper the initial $v$ field is independent of the wave type and is therefore identical to the one in Figure 4. As

in Figure 4, the patterns in the latitude-time diagrams of the simulated solutions are noticeably out of phase, but the dominant slope in the time-longitude diagrams (bottom row) agrees well with the analytic slope, indicating that the observed phase shift results from a small phase speed error that accumulates over time.

Finally, the structure-error in Figure 6 fluctuates about a mean value of less than $1\%$ and there are no visible trends throughout the 100 wave-period simulations. Recall that the structure-error defined in (8) is insensitive to phase differences.

## 4.3   Smoothness and stability

In this section we examine the generation of small-scale features and the stability of the proposed wave solutions in a similar fashion to that used in Thuburn and Li (2000) for the original Rossby-Haurwitz wave-number 4.

In Thuburn and Li (2000), the generation of small-scale features and the potential enstrophy cascade is observed by examining the potential vorticity field, which generates tongues that wrap up around themselves and break the initial east-west

symmetry. For the small wave-amplitude $A = 10^{-5}$ m s$^{-1}$ used in the present work, the potential vorticity is dominated by the planetary vorticity which is 5-6 orders of magnitudes (depending on the wave) larger than the relative vorticity. Thus, instead of the potential vorticity we examine the relative vorticity (as well as the geopotential). Figures 1-2, as well as Figures 4-5, show the evolution of these two fields between $t = 99T$ and $t = 100T$, where $T$ is the wave-period in each case. Clearly, both fields remain regular throughout the simulations and do not develop small-scale features like the ones observed in Thuburn

and Li (2000). Recall that the simulations in the present work were carried out without any diffusion/viscosity terms. Thus, the simulations remain stable for at least 100 wave-periods with no need to remove potential enstrophy at the grid scale.

In order to examine the stability of the chosen initial waves we repeat the simulations of the previous section with an added perturbation (white noise) to the initial fields. We demonstrate the stability of the waves using only the global-scale model, which was found to yield more stable results when adding the perturbation.

Figures 7 and 8 show the initial (top) fields of the perturbed Rossby and EIG waves, respectively, and the resulting latitude-time (middle row) and time-longitude (bottom row) Hovmöller diagrams of the simulated solution, obtained using GFDL's global-scale spectral model. The initial perturbation in these figures consist of a uniformly distributed random white noise with amplitude of $5\%$ of the field's amplitude added to each of the fields $u, v, \Phi$. Specifically, let $q$ stands for any of the variables $u, v$

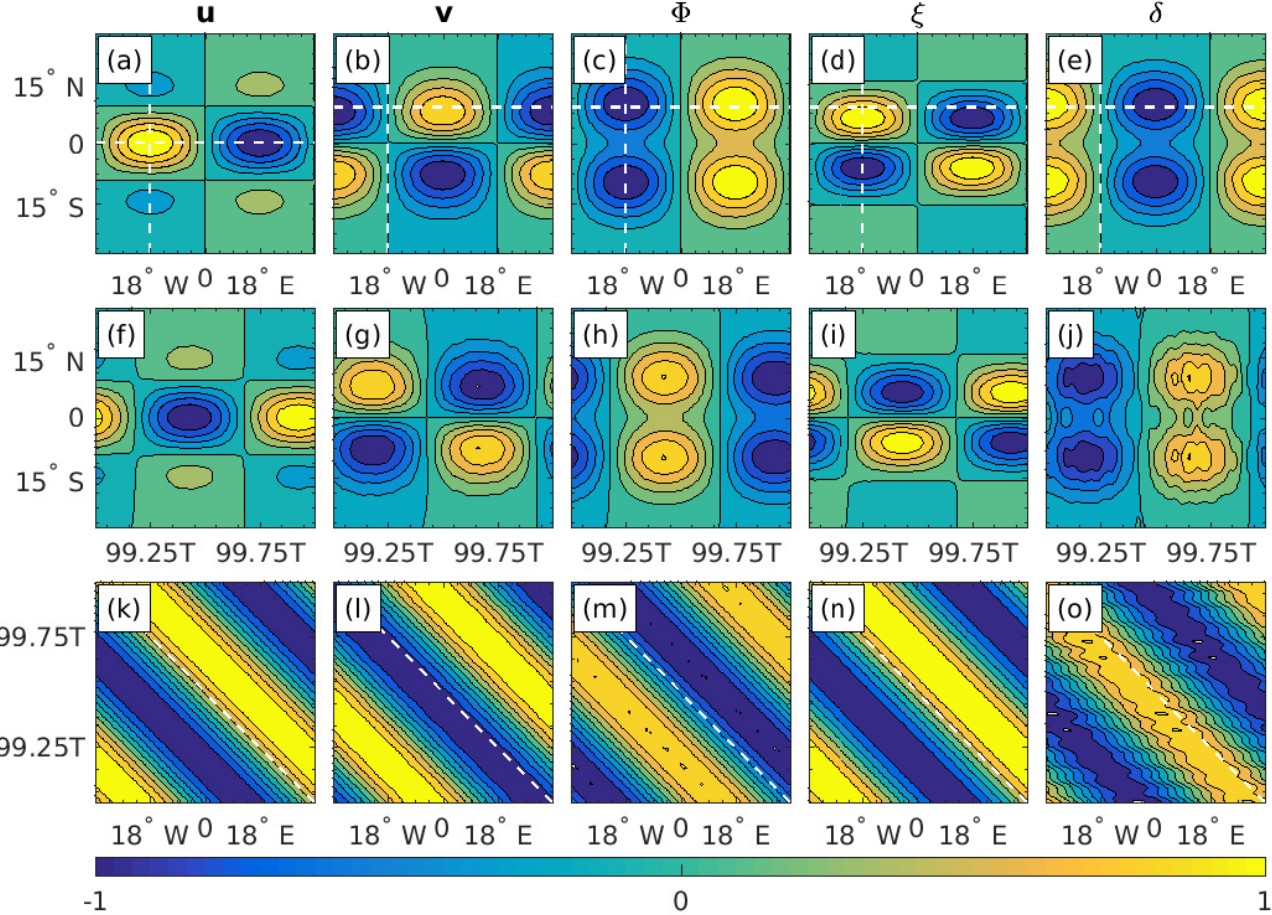

**Figure 4.** Top row: the initial $u, v, \Phi, \xi, \delta$ Rossby wave fields (top row), obtained using the analytic expressions of Section 2 and wave-parameters of Section 3.1. Middle row: latitude-time Hovmöller diagrams of the simulated solutions, obtained by intersecting the fields at $\lambda = -18°$ (also indicated by white vertical dashes lines in the top row). Bottom row: time-longitude Hovmöller diagrams of the simulated solutions, obtained by intersecting $u$ at $\phi = 0°$ and all other fields at $\phi = 9°$ (also indicated by white horizontal dashes lines in the top row). The simulated solutions were obtained using GFDL's global-scale spectral model. The fields are normalized on their global maximum at $t = 0$. The wave-period for the chosen wave-parameters is $T = 18.5$ days. Contour-levels range from $-1.0$ to $+1.0$ by $0.2$.

or $\Phi$ then the initial perturbation is given by

$$q = q_a + 0.05 \max_{\lambda, \phi} |q_a|(2R - 1), \tag{11}$$

where $q_a$ is the analytic solutions obtained as in Section 2, and $R$ is a uniformly sampled random matrix with elements in $(0, 1)$ whose dimensions are the same as $q_a$ (in the present work a different $R$ was drawn for each of the three variables).

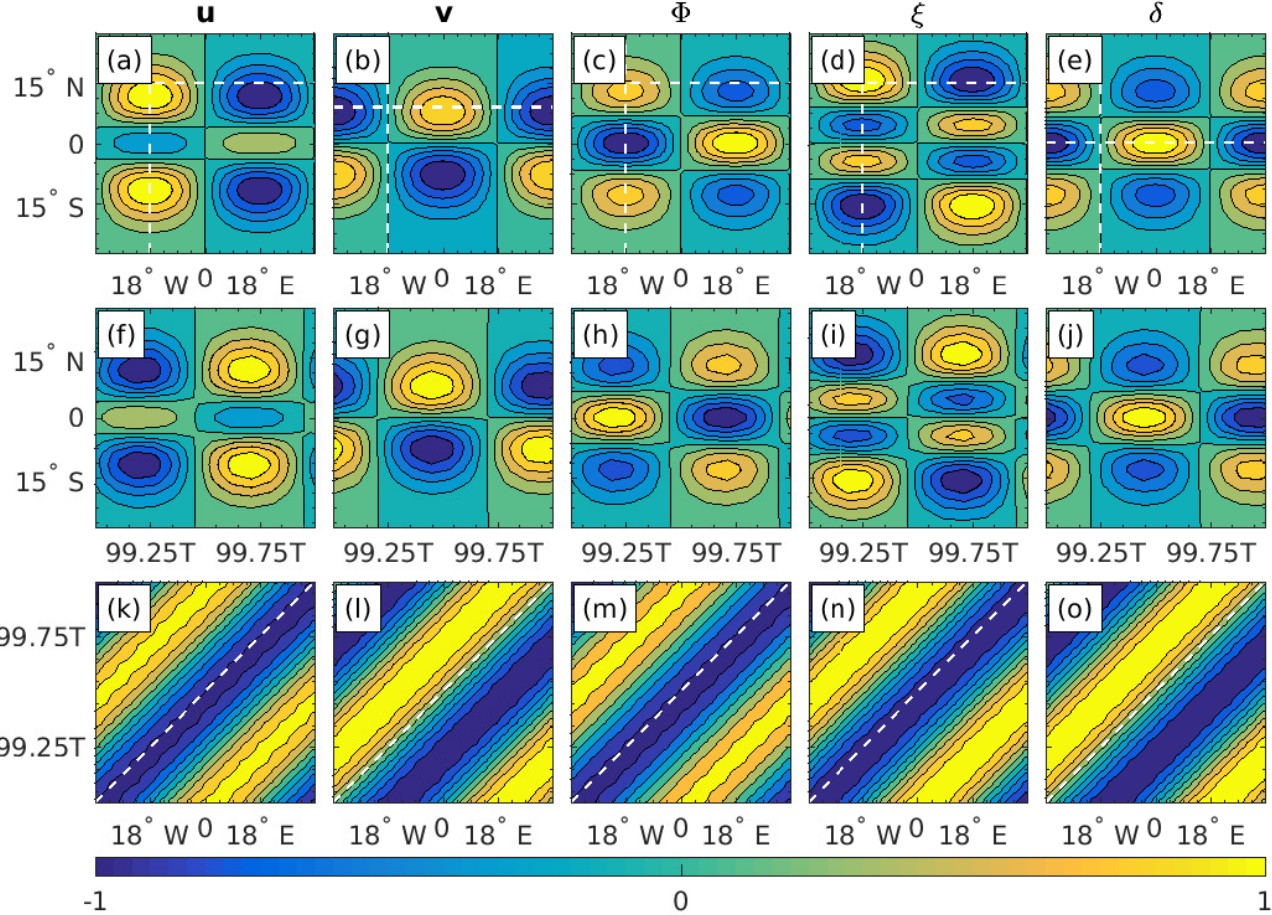

**Figure 5.** Top row: the initial $u, v, \Phi, \xi, \delta$ EIG wave fields (top row), obtained using the analytic expressions of Section 2 and wave-parameters of Section 3.1. Middle row: latitude-time Hovmöller diagrams of the simulated solutions, obtained by intersecting the fields at $\lambda = -18°$ (also indicated by white vertical dashes lines in the top row). Bottom row: time-longitude Hovmöller diagrams of the simulated solutions, obtained by intersecting $v$ at $\phi = 9°$, $\delta$ at $\phi = 0°$ and all other fields at $\phi = 15°$ (also indicated by white horizontal dashes lines in the top row). The simulated solutions were obtained using GFDL's global-scale spectral model. The fields are normalized on their global maximum at $t = 0$. The wave-period for the chosen wave-parameters is $T = 1.9$ days. Contour-levels range from $-1.0$ to $+1.0$ by $0.2$.

Overall, the perturbed waves seem to be stable. The $u,v$ and $\Phi$ fields are almost as regular as those of the non-perturbed waves, except for the zero-contour. The small-scale features in the vorticity field of the perturbed Rossby smooth out with time, in contrast to the potential vorticity field of the Rossby-Haurwitz wave-number 4. On the other hand, the perturbed Rossby wave divergence field is completely eroded. The vorticity and divergence fields of the perturbed EIG wave are not as regular as those of the non-perturbed wave. However, they too become smoother with time and the initial wave remains the most

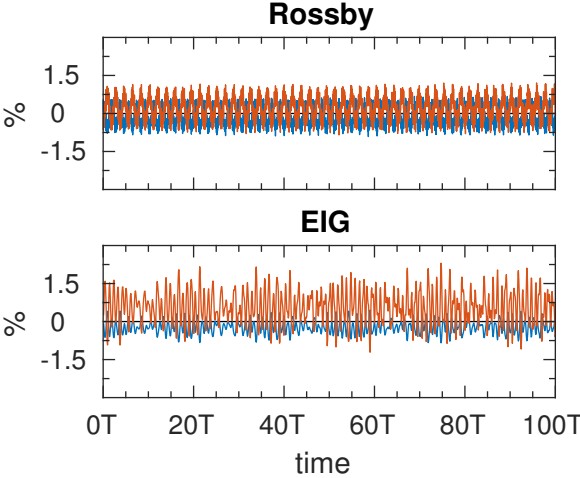

**Figure 6.** The structure-error defined in (8) for both the Rossby (top) and EIG (bottom) waves as a function of time. Blue: calculated for the velocity vector $\mathbf{u} = (u, v)$. Red: calculated for the geopotential $\Phi$.

dominant wave throughout the entire 100 wave-period simulation. The structure-error in Figure 9 is similar to the previous ones in Figures 3 and 6. These results are quite surprising. We would expect a sufficiently large perturbation to excite other modes, regardless of the waves' stability.

Both the non-perturbed Rossby wave in Sections 4.1 and 4.2, and the perturbed Rossby wave of the present section indicate that the divergence field is more sensitive than the other four fields of the Rossby wave. An immediate suspect in this regard is the divergence field amplitude, which is small for the chosen Rossby wave. For reference the meridional wind amplitude for the chosen waves parameters (of both the Rossby and EIG waves) is $6.4e - 6$, whereas the Rossby wave divergence field amplitude is $2.6e - 12$. On the other hand, the divergence field amplitude is only one order of magnitude smaller than the vorticity field amplitude, which is $2.7e - 11$. Regardless of the cause, the fact that all other four fields remain quite regular while the divergence field is completely eroded suggests that the small-but-significant divergence field described by Phillips (1959) is in fact a small-and-insignificant one.

## 4.4 Convergence test for the linear shallow water models

In addition to the test cases proposed by Williamson et al. (1992) a resolution convergence test of linearized SWEs in which the simulations are compared to higher order simulations is also useful for ensuring that the errors decrease with the increase in resolution. In this section we demonstrate that Matsuno's analytic wave solutions can be used for this purpose. We use the equatorial channel model which can be easily turned into a linear shallow water model.

Figure 10 shows the structure-error in absolute value as a function of the grid-spacing $\Delta = \Delta\lambda = \Delta\phi$, from $\Delta = 2.5°$ to $\Delta = 0.5°$ every $0.25°$. For each resolution, the initial non-perturbed waves were integrated for 100 wave-periods. As an estimate of the structure-error at each resolution we use the time-series averages (indicated by dots). The error-bars were

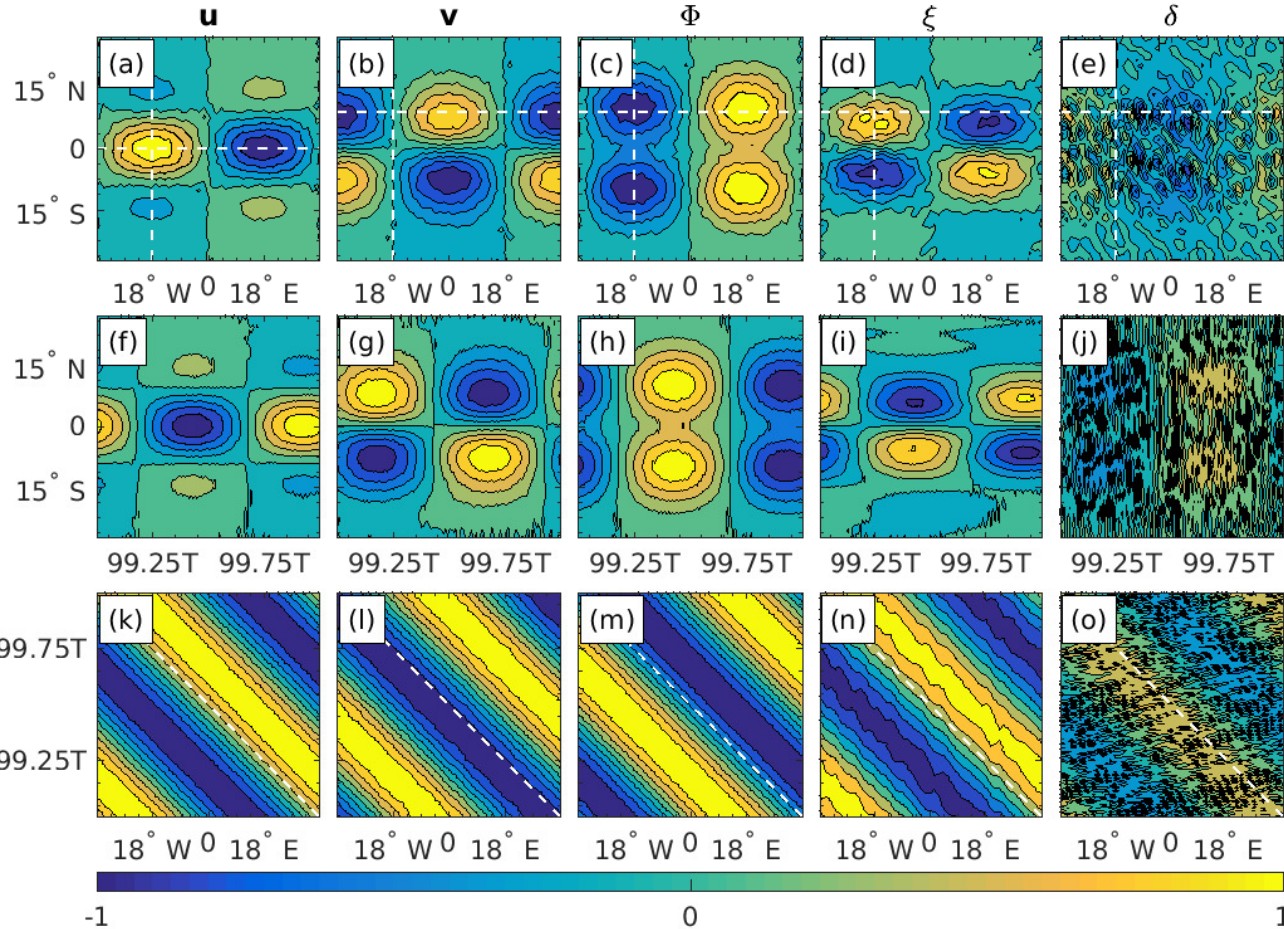

**Figure 7.** Same as Figure 4, but for the perturbed Rossby wave.

estimated using the standard deviations of the entire time-series. As the resolution increase from $\Delta = 2.5°$ to $\Delta = 0.5°$, the structure-error time-series average decrease from about about $2\%$ to less than $1\%$, while the standard deviation decrease from about $2\%$ to about $0.5\%$. The time step across all resolutions in this figure was held fixed at $\Delta t = 100$ sec. Note that all the results of the previous sections were obtained for $\Delta = 0.5°$ and $\Delta t = 600$ sec. This time step was found to yield convergent results for $\Delta = 0.5°$ in the sense that decreasing the time step by a factor of two yields no improvements. Nevertheless, for the convergence test of the present section we have further decreased the time step to $\Delta t = 100$ sec in order to allow further increase in the spatial resolution by another $0.25°$. This has also enabled a comparison with the results of the previous sections, thus ensuring that the simulations remain stable. Needless to say, for any time step one can expect to encounter numerical instabilities at some (high) spatial resolution.

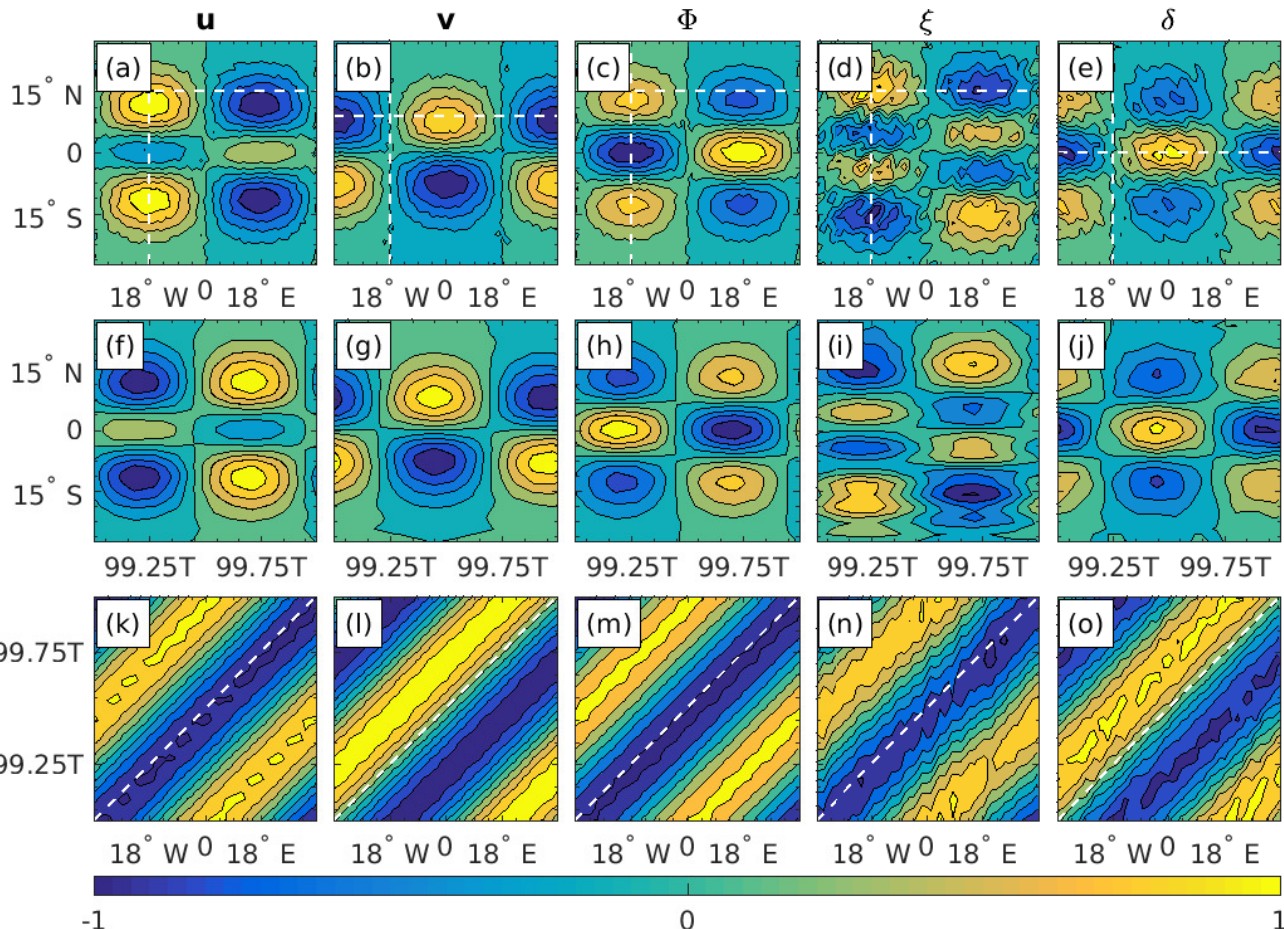

**Figure 8.** Same as Figure 5, but for the perturbed EIG wave.

## 5 Concluding remarks

As vertical resolutions in atmospheric and oceanic models increase it is essential to assess the accuracy with which they resolve baroclinic wave modes, typified by small gravity wave phase speed, in addition to the barotropic mode. To this end we propose to use a similar procedure to that used in the Rossby-Haurwitz test case but with different initial conditions. Instead of using the analytic solutions obtained by Haurwitz (1940), which are only accurate for large gravity wave speeds such as those of the barotropic mode, we propose to use the analytic solutions obtained by Matsuno (1966), which are accurate for smaller gravity wave speeds such as those of the baroclinic modes.

While Matsuno's solutions apply for the equatorial $\beta$-plane, they approximate the solutions of the SWEs on the sphere for the speeds of gravity waves found in the baroclinic modes in the atmosphere, and as demonstrated in the present work can be accurately simulated in both equatorial channel and global-scale models in spherical coordinates. In addition, unlike

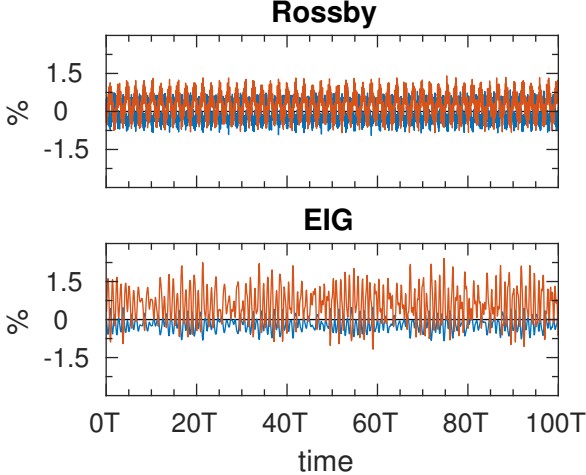

**Figure 9.** Same as Figure 6, but for the perturbed Rossby (top) and EIG (bottom) waves.

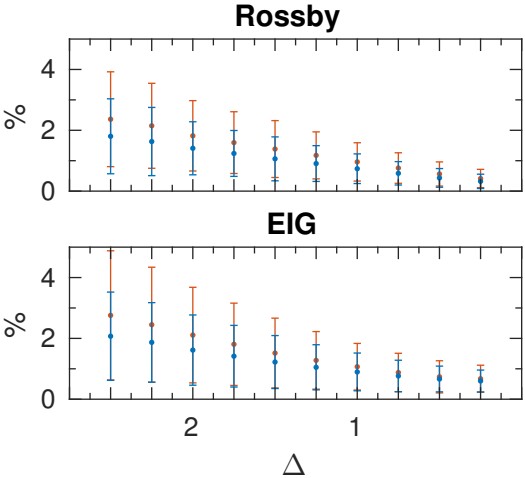

**Figure 10.** Structure-error in absolute value as a function of the grid-spacing $\Delta = \Delta\lambda = \Delta\phi$, from $\Delta = 2.5°$ to $\Delta = 0.5°$ every $0.25°$. The points correspond to the time averaged structure-error over 100 wave-periods, and the error-bars are determined from the standard deviation. Blue: calculated for the velocity vector $\mathbf{u} = (u, v)$. Red: calculated for the geopotential $\Phi$

the original Rossby-Haurwitz wave-number 4, the chosen initial waves of the present test case remain stable for at least 100 wave-periods, which for the chosen Rossby wave correspond to about 1850 days.

While the solutions of the SWEs obtained by Matsuno (1966) account for the small divergence field missing from the non-divergent Rossby-Haurwitz waves, the results of the present study suggest that this missing divergence field is insignificant.

5      Ideally, we expect the proposed test case to stand on an equal footing alongside the Rossby-Haurwitz one, but in the words of Williamson et al. (1992): "The test will only become standard to the extent that the community finds it useful".

*Code availability.* A Python module for evaluating the initial conditions and analytic solutions is publicly available under the MIT license at https://github.com/ofershamir/matsuno, and archived on Zenodo at http://doi.org/10.5281/zenodo.2605203.

*Data availability.* TEXT

*Code and data availability.* TEXT

5 *Sample availability.* TEXT

*Author contributions.* NP conceived the idea of standardizing the Matsuno test case for General Circulation Models in spherical coordinates. IY adopted the Cartesian shallow water model used in Gildor et al. (2016) to spherical coordinates and was responsible for the numerical simulations. OS analyzed the numerical results, prepared the manuscript and ran the GFDL spectral global model. SZZ prepared the IC generating code for packaging, deployment, testing and licensing.

10 *Competing interests.* The authors declare that they have no conflict of interest.

*Disclaimer.* TEXT

*Acknowledgements.* H-Z. Krugliak and Dr. C. I. Garfinkel of HU helped us install and run the GFDL model. We also acknowledge the helpful discussions we had with Dr. Y. De-Leon of HU and the insightful comments of the two anonymous reviewers of our manuscript.

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
