# Peer review of "The Matsuno baroclinic wave test case"

_Geoscientific Model Development, 2018_

## Editor Comment (EC1) · D.A. Ham (Editor) · 5 Nov 2018

I will not pass comment at this stage on the relevance or correctness of this test case, since the nominated reviewers are much more qualified than I am to assess this. However I will address the issue of the reference implementation which has been provided. The reference implementation is important in a test case manuscript, as a well-executed reference implementation will greatly increase the attractiveness of the test case to model developers. Regrettably, at this stage the reference implementation has a number of issues which make it less likely to be used.

[Figure]

**1 Deployability**

Individual source files in a paper supplement are not particularly easy for language users to employ. This is unfortunate because Python and Matlab (though not really Fortran) provide straightforward mechanisms for packaging files to make them easily installable and usable. In the case of Python, if the package files are then hosted on an online revision control system such as GitHub then users will be able to directly pip install them with a single command. Instructions on how to create a Python package are at https://packaging.python.org/tutorials/packaging-projects/ while Matlab toolboxes are at: http://www.mathworks.com/help/matlab/matlab_prog/create-and-share-custom-matlab-toolboxes.html.

**2 Verification and unit testing**

There is currently no testing of the code provided. There is little more frustrating when using a test case than to eventually discover that the bug was actually in the test case. Potential users will therefore want to see that there is a robust set of verification tests that demonstrate that the implementation is correct. In this case you have two routes which can both be used. The first is to check the output of your routines for parameter values where the output is easy to verify. The second is to cross-verify your code by calling all the implementations and checking that their output is the same. For this purpose, it is useful to note that the Matlab code might be callable from Python using https://pypi.org/project/oct2py/. It is possible to directly call Fortran from Python using f2py, or you could use the Fortran standard C interoperability features to present a public C interface (which would also be a nice user feature!) and call that using ctypes. There is some documentation on how to do this in the https://docs.scipy.org/doc/numpy-1.15.4/user/c-info.python-as-glue.html.
You should use one or more test frameworks (maybe one, maybe you need one for each language) to run the testing in a way that users will understand and which is easy for them to integrate into their own testing frameworks. For example you might use https://pytest.readthedocs.io/en/latest/. There are likewise suitable testing frameworks available for Matlab and Fortran (or, if you set up cross-verification, you could just use the Python framework for all languages).

**3 Code interfaces and naming**

As far as I can see, the public interface of the code consists primarily of the function `get_fields`, which seems reasonable given that the purpose of the code is to execute one particular test case. However there are a number of things about this interface which are unhelpful. First, the name `get_fields` is problematic. It's not going to be obvious to the user that that is the main function and even what it does. A more descriptive name would be useful, potentially without the `get_` prefix, which adds no information. In Python, the auxiliary routines which are not supposed to be visible outside the package should be prefixed with underscore to mark them as private.

In both Matlab and Python, the documentation for functions should be placed in the docstring at the start of the function, not at the start of the file. This change will enable the self-documentation features of these languages to work. It should also be borne in mind that users may see the code without the paper, so it would be a good idea if the docstrings referred back to the paper.

The code returns values on a regular latitude-longitude grid. This is not useful for many users. Even many structured grid models have curvilinear grids which are not axis-aligned, and this is before considering unstructured mesh models. The functions should instead evaluate the results at an arbitrary set of latitude-longitude points. The modest performance gain that may be available from the current structure is not worth

the cost of having code that many users simply will not be able to use.

It is quite conceivable that the user will want to change $g$, $\Omega$, $A$, or $H_0$ in order to run the test case for a model using scaled parameters. Currently in Fortran these are private, and in all languages this change would involve changing the global state of the module, which is an unsafe programming practice. It would be better if these parameters were optional arguments to the main function(s), with default values being those given currently.

The test case is only valid for certain values of input parameters. In addition to documenting these limitations, the code should check for illegal values and raise appropriate exceptions or return some error, depending on language.

**4  Licence**

There is currently no licence provided with the code. This might mean that the code falls under the same Creative Commons attribution licence as the paper, but since the code may be separated from the paper when used, it would be preferable to provide an explicit licence. Many authors would use the MIT License in this context, which basically says that users can do anything they like so long as they continue to acknowledge the authorship of the code. See https://opensource.org/licenses/MIT. In any event, an explicit licence should be chosen and included with the source files.

**5  GitHub and archiving**

The steps above will result in quite a lot more files in the implementation, along with a nontrivial directory structure. This could be attached in the supplement, but this has a number of disadvantages. First, there is no mechanism for future bug fixes. Next,

direct access mechanisms such as Python's package installer pip will not see the supplement. Instead, the code could be posted on GitHub, which would fix the update and pip problems. GitHub is not a suitable archive location for the canonical version of the code in a manuscript, but GitHub and Zenodo together provide a straightforward way of archiving the repository with a DOI for inclusion in the manuscript. The instructions are at https://guides.github.com/activities/citable-code/. GitHub + Zenodo would be a better alternative to the supplement.

---

## Author Comment (AC1) · 6 Nov 2018

The, mainly technical, comments raised by the Editor will be handled when the revised version of the paper is prepared following the reviews the MS receives. It should be stressed that the proposed test case is based on explicit expressions (derived from analytic solutions of the differential equations) so that the codes are verified by the success of the simulation. An error in generating the initial fields is bound to result in an incorrect simulation in which the simulated fields deviate from the initial fields substantially after several time steps. Also, we've verified that our codes can be applied as is to arbitrary lon-lat grids and are not restricted to the regular, equally-spaced, lon-lat grid used in the MS. This is demonstrated in the attached figure that shows the initial $u$ (left), $v$ (center) and $\Phi$ (right, the geopotential!) Rossby wave fields. Top row: regular grid with a fixed 1 degree spacing in both $\lambda$ and $\phi$

[Figure]

[Figure]

[Figure]

**Fig. 1.**

u    v    Φ

(a)    (b)    (c)

5° N
0
5° S

(d)    (e)    (f)

5° N
0
5° S

15° W 0 15° E    15° W 0 15° E    15° W 0 15° E

-1    0    1

---

## Referee Comment (RC1) · Anonymous Referee #1 · 4 Dec 2018

The aim of this paper is to present a standardised test case for shallow water models which is derived from analytic solutions of the linearised shallow water equations on the equatorial beta plane published by Matsuno in 1966. Recent work by one of the authors and others has shown that solutions to these equations approximate the solutions of the full shallow water equations on the sphere in the asymptotic limit of small gravity wave speed. The authors propose 2 sets of parameters, one relevant to the atmosphere and one relevant to the first baroclinic mode of the ocean. The wavenumber and mode are chosen to be small (5 and 1 respectively) to ensure the waves are close to the asymptotic limit and so that they can be spatially resolved without requiring too high resolution. The wave amplitude is also necessarily small. For each set of parameters the authors suggest 2 simulations: one of a westward propagating Rossby wave and the other an eastward propagating inertia-gravity wave. To assess the accuracy of the simulation the authors suggest 2 measures: Hovmoeller diagrams and spectral

analysis. They present these for simulations run using a finite difference model in an equatorial channel in spherical coordinates. This paper is a continuation of the work in Shamir and Paldor (2016) which deals with the case of large gravity wave speeds.

The Rossby-Haurwitz wave described in test case 6 of Williamson et. al. (1992) is known to be problematic. Thuburn and Li (2000) describes these issues and I think that paper should be referenced here, as it was in the previous paper (Shamir and Paldor, 2016). One issue is that the original initial conditions as specified in Williamson et. al. (1992) lead to wrapping up of potential vorticity contours and the associated generation of small scale features and potential enstrophy cascade. A figure showing the potential vorticity at several times throughout the simulations would be appreciated here to show that this does not happen for this test case. The other issue with the original test case is that it is unstable. This is demonstrated numerically in Thuburn and Li (2000) by adding some small noise to the initial conditions after they noticed that the solution they computed using a finite volume model on a grid of hexagons and pentagons (i.e. their only non latitude-longitude model) broke down. The errors related to the structure of the underlying grid triggered the dynamical instability. The solutions in this paper have been computed using a regular latitude-longitude grid so I wonder if a similar issue could occur with this test case. I suggest that the authors could check this by either adding some noise to the initial conditions, as in Thuburn and Li (2000), or by running their code on a rotated grid (i.e. with the poles in the midlatitudes).

Overall I recommend this paper for publication, subject to minimal correction and clarification as described below. It is well written and clearly describes useful diagnostics to be used as error measures. However, I feel that the case for the usefulness of the proposed test case could be made stronger. For example, some papers use the Rossby wave test as a convergence test, using a reference solution from either a higher resolution run or from a different model. Could the analytical solution here be used to test the convergence of a linear shallow water model? This would provide a useful test in between the steady state test case 2 and the other tests that require a reference solution
from a higher resolution run. Also, if the wave is indeed stable it would be a fantastic replacement for test case 6, especially for unstructured grid models, or models that use adaptive mesh refinement, since truncation errors related to mesh topology will have no dynamic instability to trigger.

**Specific Comments:**

pg 4, lines 14-15: I am concerned that different pre-factors lead to less stable solutions - it makes me wonder if the version chosen in this paper is indeed stable to differences in grid alignment.

Figure 2: Is there any reason why the Rossby wave with H=0.5 is less regular than the other solutions? Why do some of the contour plots have white regions when the values have been normalised so should lie in the range [-1, 1]?

Power spectra: Are these at all sensitive to the sampling frequency? My experience is that the spectra can be very sensitive to this but maybe that is for more turbulent simulations.

Supplement: The code provided to compute the initial conditions, while appreciated, could be improved. The authors state that the code will compute the analytic fields on arbitrary latitude-longitude grids but they have assumed that these grids are structured. These codes will not work as written for unstructured meshes, which are becoming more common in the community. The test case is much more likely to be used if these codes could be amended (i.e. they return values given a list of latitude-longitude values). In addition to this, there are some unnecessarily confusing aspects of the code. For example, there is no need to capitalise variable names so the radius of the Earth, which is called $a$ in the paper could be $a$ rather than $A$ in the code. The is especially confusing since there is also an $A$ in the equations described in the paper. It would also make sense to have $H$ as an input parameter, since this can be varied.

**Technical Corrections:**

Equation 3b: This is different to that in the code matsuno.py (and I think the code is correct).

Equation 3c: I think this is missing a sqrt around the gH.

---

## Author Comment (AC2) · 10 Dec 2018

We thank Reviewer #1 for his/her insightful comments. We'll be happy to implement these comments in the revised version of the paper which we are planning to submit after we receive the second reviewer.

---

## Referee Comment (RC2) · Anonymous Referee #2 · 31 Dec 2018

Review of the GMDD manuscript 'The Matsuno baroclinic wave test case'
Authors: Ofer Shamir, Itamar Yavoby, and Nathan Paldor

**Summary:**

The manuscript introduces a test case for shallow water models that is built upon analytic solutions of the linearized shallow water equations on the equatorial beta-plane, originally published by Matsuno (1966). The new test case has some similarity to the Rossby-Haurwitz test case, a standard test case for spherical shallow water models and a solution of the non-divergent barotropic vorticity equation. However, the Rossby-Haurwitz wave becomes unstable over longer integration periods in shallow water models, as e.g. shown in Thuburn and Li (2000), and its default parameter settings lead to high-speed gravity waves. The new test case approximates the solutions of the shallow water equations for small-speed gravity waves, and therefore mimics the baroclinic modes of the atmosphere and oceans. The flow pattern of the new test case is also more stable than the barotropic Rossby-Haurwitz flow pattern. It might therefore present an interesting addition to the suite of shallow water test cases on the sphere.
Example solutions for the test case are explored for westward-traveling Rossby waves (planetary waves) and eastward-propagating inertia-gravity waves (EIG) for both atmospheric and oceanic conditions with mean shallow water heights of $H = 30$ m (atmosphere) and $H = 0.5$ m (ocean). These example solutions are generated with a tropical channel version of a shallow-water model in spherical coordinates. The analysis is based on Hovmoeller diagrams and temporal and spatial spectra. Fortran, Matlab and Python files are provided as supplemental material that can be used to compute the initial conditions and analytic solution at any time and any location.

**General comments:**
In general, the manuscript is very well written, informative and a very valuable addition to the suite of shallow-water test cases. However, some questions and corrections need to be addressed before it can be recommended for publication.

1) The manuscript claims (e.g. at the bottom of page 4) that this test case can be used for tropical-channel shallow water models (as presented in this manuscript) and global-scale models. From the manuscript it is not entirely clear that the test will work for global models due to the use of the equatorial beta-plane approximation in the derivation for e.g. the transformations of (x,y) and the wavenumber k. The modeling community (as a 'customer' of this test case) generally works with global shallow water models and tropical-channel model in spherical geometry are extremely rare. It therefore would have been more valuable (or convincing) to present example solutions for a global shallow water model instead of a tropical-channel model. Can the tropical-channel shallow water model also be configured as a global model to demonstrate that the test case works for the whole sphere? Please provide extended explanations or ideally results from a global model.

2) Model developments with regular latitude-longitude grids have become very rare over the last decade. More typical grids are now cubed-sphere, hexagonal or icosahedral grid with built-in grid irregularities. The manuscript states that the solutions of this test case are very stable for at least 10 wave periods, which is demonstrated on a regular lat-lon grid. This triggers the question whether this statement will hold for today's models with non-latitude-longitude grids. Another question is whether small perturbations of the initial conditions will disrupt or shorten the

stability of the test case. Please provide information on these aspects.

3) As detailed below (points 5-7), the description of the initial conditions is incomplete. In addition, the analytic equations (Eq.(3)) differ slightly from the implementation in the Fortran, Matlab and Python codes. The test is therefore not usable by others in its current form, and the manuscript/codes need to be corrected.

**Technical comments:**
1) Page 1, line 9, also page 2, line 32: Please describe the model as an 'equatorial channel' model.
2) Page 1 line 12, page 2 line 2, page 5 lines 3&7: Generalize the description of the grids. A test case for only 'latitude-longitude' grids will have rather limited use. I think you meant to say that given the location of a latitude and longitude, the initial conditions and analytic solutions can be computed on any grid.
3) Page 2, line 24: It is incorrect to say that the term 'baroclinic' is associated with density variations in the vertical directions. A flow with identical density and pressure variations (e.g. for isothermal conditions) is still barotropic. Density and pressure variations need to vary independently of each other.
4) Page 3, line 15: What is meant by 'reduced gravity'. The initialization of the test case uses the regular Earth's gravity. Modify.
5) Page 3, line 17, also page 5&6 section 3.1: The wave mode n=1 is selected which leads to three distinct real roots in Eq. (2). Two of these roots are selected for the example results, but no equations are given for the Rossby wave root and EIG root. Without this information, the description of the initial conditions is incomplete. Add this information to Section 3.1.
6) Page 4, Eq. (3) and text: The manuscript fails to explain the meaning and definition of $\psi_n$. What is the relationship between $\psi_n$ and the normalized Hermite polynomials $H_n$? Without the definition of $\psi_n$ the description of the initial conditions is incomplete.
7) Page 4, Eq. (3): Eq. (3) seems to be correct, but the Fortran/Matlab.Python scripts use a wrong u_hat calculation. E.g. the Fortran code in line 200 needs to read sqrt(g*H0) instead of just 'g*H0'.
8) Page 4, line 10: State that the amplitude A needs to have units of m/s.
9) Page 4, line 25: you imply that the planar wavenumber k is unitless, so that that spherical wavenumber $k/(a \cos\phi_0)$ has units of 1/m. Please comment and clarify. Correct typo, should be 'replaced'.
10) Page 8, line 1: What is meant by the 'transport form' of the SWEs? This seems to imply the 'advective form'. However, the provided equations are in 'conservation form'.
11) Page 8, line 10: Explicitly state whether the example model uses diffusion or smoothing/filtering operations for the computations, and if yes, which ones. Should users of the test case try to omit all diffusion/filtering operations in their models when using this test case? E.g. the provided shallow water code contains provisions for a temporal Asselin filter.
12) Page 9, Fig. 2: The value for the symbol $\phi_f$ is not provided. Add this information.
13) Page 9, Fig. 2: It is highly unusual and confusing to see and interpret the flipped Hovmoeller diagrams. Typically, Hovmoeller diagrams list the position along the x-axis and time along the y-axis. I recommend flipping the axes in Fig. 2 to make the interpretation of the Hovmoeller diagrams easier.
14) Supplemental material: Please add Fortran/Matlab wrapper codes that will enable the user to

create/test the initial conditions. In addition, the codes should not expect to receive regular longitude and latitude arrays, but should be callable for any longitude and latitude position.

---

## Author Comment (AC3) · 8 Jan 2019

**General response**

We were happy to see that the reviewers found the proposed test case useful for model developers. We appreciate the reviewers' comments and found their suggestions helpful in making our proposed test case even more useful. We address all their comments below and have already implemented most of their suggestions in a revised manuscript. Having said that, we are unable to meet the journal's standards with regard to the supplied codes (e.g. "reference implementation") at this time. Our original intent in submitting the codes for computing the initial conditions was to provide a "first aid" rather than a "commercial grade" software package. The code essentially evaluates algebraic relations that describe the initial (u, v, Φ) fields and requires about 150 lines of code to do so. Evidently, the developers can write the required code better than us and do it in a way that better suits their particular needs, e.g. unstructured meshes. Therefore, we will not include the Fortran and Python codes in the assets of the paper and leave the Matlab code only for the sake of reproducibility. We will accept the editor's decision on this matter. In light of the referees' overall favorable reviews, and our efforts to accommodate their suggestions, we hope that manuscript is accepted for publication in GMD.

Following the reviewers' suggestions (see below), we have implemented the changes summarized below in the revised manuscript.

**Added subsections in the Results section:**
1. A demonstration of the applicability of the test case to a global-scale model (GFDL, RSW model).
2. An examination of the stability of the waves along the lines used in Thuburn and Li (2000).
3. A demonstration of the use of the proposed test case a linear convergence test (Ref. 1).

**Removed items:**
1. The H=0.5 m waves were removed. The results of the simulations with this value were less robust. In addition, adhering to a single value of H simplifies the test case.

**Replaced items:**
1. The spectral analyses were found to be too sensitive to be used for assessment purposes and were therefore replaced with the difference between the global means, which unlike the L2 norm employed by Williamson et al (1992) is insensitive to phase speed errors.

**Response (in blue) to the referees' comments (in black)**

**Anonymous referee #1**

**General comments:**

The Rossby-Haurwitz wave described in test case 6 of Williamson et. al. (1992) is known to be problematic. Thuburn and Li (2000) describes these issues and I think that paper should be referenced here, as it was in the previous paper (Shamir and Paldor, 2016).

The findings of Thuburn and Li (2000) on the Rossby-Haurwitz wave are discussed in the revised Introduction. In addition, we added a new subsection to the Results section, where we examine the generation of small-scale features and the stability of the proposed test case, similar to the way it is done in Thuburn and Li (2000).

One issue is that the original initial conditions as specified in Williamson et. al. (1992) lead to wrapping up of potential vorticity contours and the associated generation of small scale features and potential enstrophy cascade. A figure showing the potential vorticity at several times throughout the simulations would be appreciated here to show that this does not happen for this test case.

For the small wave amplitude, A=1e-5 ms$^{-1}$, used in our test case the potential vorticity is dominated by the planetary vorticity. Therefore, we added Hovmöller diagrams of the relative vorticity, instead of the potential vorticity, which show that there is no generation of small-scale features during the last wave-period of the simulation. See new section 4.3 and 4$^{th}$ columns in new Figs 1,2,4,5,7 and 8.

The other issue with the original test case is that it is unstable. This is demonstrated numerically in Thuburn and Li (2000) by adding some small noise to the initial conditions after they noticed that the solution they computed using a finite volume model on a grid of hexagons and pentagons (i.e. their only non latitude-longitude model) broke down. The errors related to the structure of the underlying grid triggered the dynamical instability. The solutions in this paper have been computed using a regular latitude-longitude grid so I wonder if a similar issue could occur with this test case. I suggest that the authors could check this by either adding some noise to the initial conditions, as in Thuburn and Li (2000), or by running their code on a rotated grid (i.e. with the poles in the midlatitudes).

Similar to Thuburn and Li (2000) we added a small (5% in our MS) uniformly distributed random noise (perturbation) to the initial conditions (IC). The simulations with the perturbed IC demonstrate that after 100 wave-periods the simulated solutions preserve the initial wave structure. In particular, the small-scale features in the initial u, v, Φ and ξ fields smooth out and do not generate smaller-scale features. See new section 4.3.

some papers use the Rossby wave test as a convergence test, using a reference solution from either a higher resolution run or from a different model. Could the analytical solution here be used to test the convergence of a linear shallow water model? This would provide a useful test in between the steady state test case 2 and the other tests that require a reference solution from a higher resolution run.

We added a convergence test of the linear shallow water model, which demonstrates that the "error" decreases "exponentially" as the resolution increases. Following one of the other comments by the

reviewer on the sensitivity of the spectral analyses we adopted a different assessment criterion which is also used to estimate the error here. See the revised section 3.2 and new section 4.4.

Also, if the wave is indeed stable it would be a fantastic replacement for test case 6, especially for unstructured grid models, or models that use adaptive mesh refinement, since truncation errors related to mesh topology will have no dynamic instability to trigger.

We hope that the revised version of the manuscript, and, in particular, the addition of noise in section 4.3, is more convincing than the previous version. We, too, view the proposed Matsuno test case as a substitute for test case 6 and hope it is adopted by the community.

**Specific comments:**

1. pg 4, lines 14-15: I am concerned that different pre-factors lead to less stable solutions - it makes me wonder if the version chosen in this paper is indeed stable to differences in grid alignment.

This is a subtle question. There is no reason why Matsuno's expressions should be more stable. We imagine that the most optimal choice of pre-factors can depend on considerations e.g. the prognostic variables used. For example, it is quite possible that different choices are more optimal for models that use vorticity-divergence. Note also that the different pre-factors originate from the use of the normalized Hermite functions whose amplitudes are bounded (Cramér's inequality), as oppose to the amplitudes of the non-normalized Hermite functions that grow indefinitely as $n$ increases. Thus, for large $n$ we expect Matsuno's expressions to be less stable numerically. On the other hand, for the chosen n=1 it is unclear whether the difference between the two forms has any effect on there dtability. Finally, while the present choice might not be the most optimal, the simulated solutions seem to be stable for 100 wave-periods.

2. Figure 2:

Is there any reason why the Rossby wave with H=0.5 is less regular than the other solutions?

We are unsure but the wave modes of H=0.5 m were deleted altogether from the revised manuscript.

Why do some of the contour plots have white regions when the values have been normalised so should lie in the range [-1, 1]?

As is stated in the figure caption, the fields are normalized on the global maximum at t=0. Therefore, the white regions correspond to times when the field's global extrema temporarily exceed the [-1,1] contour range. In our opinion normalizing on the global maximum at t=0 and keeping the contour range fixed is the better option. We added a clarification in the text in the paragraph discussing Figure 1.

3. Power spectra: Are these at all sensitive to the sampling frequency? My experience is that the spectra can be very sensitive to this but maybe that is for more turbulent simulations.

The reviewer is right. By sub-sampling our results by factors of 2 or 4 (so as to insure there are at least 2.5 samples per wave-period) it was evident that while the power spectra were generally similar, the results can be too sensitive to be used for assessment purposes. Therefore, we adopted a different assessment criterion, which is also simpler than the spectral analyses. See the revised section 3.2.

4. Supplement: The code provided to compute the initial conditions, while appreciated, could be improved. The authors state that the code will compute the analytic fields on arbitrary latitude-longitude grids but they have assumed that these grids are structured. These codes will not work as written for unstructured meshes, which are becoming more common in the community. The test case is much more likely to be used if these codes could be amended (i.e. they return values given a list of latitude-longitude values). In addition to this, there are some unnecessarily confusing aspects of the code. For example, there is no need to capitalise variable names so the radius of the Earth, which is called

a in the paper could be a rather than A in the code. The is especially confusing since there is also an A in the equations described in the paper. It would also make sense to have H as an input parameter, since this can be varied.

Thank you, but we have decided to leave the computation of the initial conditions to the developers that can do it better than us and do it in a way that suits their particular needs, e.g. unstructured meshes.

**Technical corrections:**

1. Equation 3b: This is different to that in the code matsuno.py (and I think the code is correct).

Equation 3b and the code are consistent and both are correct! Note that, in the Fortran code for example, in addition to the different pre-factor in line 200, the expressions in lines 193-194 are also different from the text. The expressions in the code are obtained from the ones in the text by taking another $(gH)^{0.5}$ factor outside of the square brackets, so that the pre-factors of $\hat{u}$ and $\hat{\Phi}$ both have $(gH)$ in the numerator, but $\omega$ in 3b is divided by $(gH)^{0.5}$. This was also flagged by Referee #2. Clearly, the difference between the code and the text is confusing. In the revised manuscript we change 3b to match the expressions in the codes.

2. Equation 3c: I think this is missing a sqrt around the gH.

Again, Equation 3c is correct! Dimensional consideration suggests that the referee's suggestion cannot be correct.

With regard to the last two comments, we have repeated the derivation of the expressions in Equation 3 from scratch and derived the same expressions as in the previous version. Also, we encourage the community to implement the test, including different pre-factors and/or different powers of (gH).

**Anonymous referee #2**

**General comments:**

1. The manuscript claims (e.g. at the bottom of page 4) that this test case can be used for tropical-channel shallow water models (as presented in this manuscript) and global-scale models. From the manuscript it is not entirely clear that the test will work for global models due to the use of the equatorial beta-plane approximation in the derivation for e.g. the transformations of (x,y) and the wavenumber k. The modeling community (as a 'customer' of this test case) generally works with global shallow water models and tropical-channel model in spherical geometry are extremely rare. It therefore would have been more valuable (or convincing) to present example solutions for a global shallow water model instead of a tropical-channel model. Can the tropical-channel shallow water model also be configured as a global model to demonstrate that the test case works for the whole sphere? Please provide extended explanations or ideally results from a global model.

We added a new subsection to the Results section where we repeat the simulations using a global-scale model (the GFDL, RSW Model). The equatorial channel model cannot be easily adopted to the entire sphere due to the convergence of longitudinal lines at the poles. Therefore, we used GFDL's global-scale model which is spectral. Please see the new section 4.2.

2. Model developments with regular latitude-longitude grids have become very rare over the last decade. More typical grids are now cubed-sphere, hexagonal or icosahedral grid with built-in grid irregularities. The manuscript states that the solutions of this test case are very stable for at least 10 wave periods, which is demonstrated on a regular lat-lon grid. This triggers the question whether this statement will hold for today's models with non-latitude-longitude grids. Another question is whether small perturbations of the initial conditions will disrupt or shorten the stability of the test case. Please provide information on these aspects.

Unfortunately, we are unable to provide results with a non-latitude-longitude grid model. We hope the community will employ the Matsuno test case with such models and comment on the subject.

With regard to the perturbations, we added a new subsection to the Results section where we examine the stability of the chosen waves. As in Thuburn and Li (2000) we added a small (5% in our MS) uniformly distributed random noise (perturbation) to the initial conditions (IC). The simulations with the perturbed IC demonstrate that after 100 wave-periods the simulated solutions preserve the initial wave structure. In particular, the small-scale features in the initial u, v, $\Phi$ and $\xi$ fields smooth out and do not generate smaller-scale features. See new section 4.3.

3. As detailed below (points 5-7), the description of the initial conditions is incomplete. In addition, the analytic equations (Eq.(3)) differ slightly from the implementation in the Fortran, Matlab and Python codes. The test is therefore not usable by others in its current form, and the manuscript/codes need to be corrected.

All the required information can be found in the original manuscript, and Equation (3) and the code are consistent and are both correct! Evidently, the original version was not clear/organized enough. We hope that the revised version does a better job at conveying the information. Please see detailed response to points 5-7 below.

**Technical comments:**

1. Page 1, line 9, also page 2, line 32: Please describe the model as an 'equatorial channel' model.

We now refer to the model as an 'equatorial channel' model as requested.

2. Page 1 line 12, page 2 line 2, page 5 lines 3&7: Generalize the description of the grids. A test case for only 'latitude-longitude' grids will have rather limited use. I think you meant to say that given the location of a latitude and longitude, the initial conditions and analytic solutions can be computed on any grid.

Fixed

3. Page 2, line 24: It is incorrect to say that the term 'baroclinic' is associated with density variations in the vertical directions. A flow with identical density and pressure variations (e.g. for isothermal conditions) is still barotropic. Density and pressure variations need to vary independently of each other.

Rephrased in the revised version

4. Page 3, line 15: What is meant by 'reduced gravity'. The initialization of the test case uses the regular Earth's gravity. Modify.

The reviewer is right. As stated in page 5, line 24 of the original manuscript, we control the speed of gravity waves $(gH)^{0.5}$ by holding $g$ fixed and equal to the Earth gravitational acceleration and varying $H$. The use of the term 'reduced gravity' originates from the fact that the linearized shallow water equations can also be derived as the horizontal structure equations in a stratified layer (in the linear case with a motionless mean flow), in which case Earth gravity is replaced by the reduced gravity and

the layer depth by the equivalent height. In order to avoid confusion, we removed these two terms and in the revised manuscript we now adhere to a "single layer" fluid.

5. Page 3, line 17, also page 5&6 section 3.1: The wave mode n=1 is selected which leads to three distinct real roots in Eq. (2). Two of these roots are selected for the example results, but no equations are given for the Rossby wave root and EIG root. Without this information, the description of the initial conditions is incomplete. Add this information to Section 3.1.

This information was provided in Appendix A of the original manuscript. In the revised version this information is moved to the main text after Equation (2) in Section 2, which is more suitable than sec. 3.1.

6. Page 4, Eq. (3) and text: The manuscript fails to explain the meaning and definition of $\psi_n$.
What is the relationship between $\psi_n$ and the normalized Hermite polynomials $H_n$? Without the definition of $\psi_n$ the description of the initial conditions is incomplete.

$\psi_n$ equals $\hat{v}_n$. Thus, in the revised manuscript we have decided to remove $\psi_n$ altogether and adhere to $\hat{v}_n$, which is just the latitude-dependent amplitude of the meridional velocity.

7. Page 4, Eq. (3): Eq. (3) seems to be correct, but the Fortran/Matlab.Python scripts use a wrong u_hat calculation. E.g. the Fortran code in line 200 needs to read sqrt(g*H0) instead of just 'g*H0'.

Equation 3 and the code are consistent and both are correct! Note that, in the Fortran code for example, in addition to the different pre-factor in line 200, the expressions in lines 193-194 are also different from the text. The expressions in the code are obtained from the ones in the text by taking another $(gH)^{0.5}$ factor outside of the square brackets, so that the pre-factors of $\hat{u}$ and $\hat{\Phi}$ both have $(gH)$ in the numerator, but $\omega$ in 3b is divided by $(gH)^{0.5}$. This was also flagged by Referee #1. Clearly, the difference between the code and the text is confusing. In the revised manuscript we change 3b to match the expressions in the codes.
We have repeated the derivation of the expressions in Equation 3 from scratch and derived the same expressions as in the previous version. Also, we encourage the community to implement the test, including different pre-factors and/or different powers of $(gH)$.

8. Page 4, line 10: State that the amplitude A needs to have units of m/s.

Added – Thank you

9. Page 4, line 25: you imply that the planar wavenumber k is unitless, so that that spherical wavenumber k/(a cos$\varphi_0$) has units of 1/m. Please comment and clarify.

The planar wave-number has units of 1/length, while the spherical wave-number is dimensionless. To avoid any confusion we added a subscript 's' for spherical variable and a comment in the text.

Correct typo, should be 'replaced'. Corrected. Thank you

10. Page 8, line 1: What is meant by the 'transport form' of the SWEs? This seems to imply the 'advective form'. However, the provided equations are in 'conservation form'.

The reviewer is right, the equations are in 'conservation form' - corrected.

11. Page 8, line 10: Explicitly state whether the example model uses diffusion or smoothing/filtering operations for the computations, and if yes, which ones. Should users of the test case try to omit all diffusion/filtering operations in their models when using this test case? E.g. the provided shallow water code contains provisions for a temporal Asselin filter.

The equatorial channel model has no diffusion/viscosity terms. It does contain provisions for a Robert-Asselin filter, but in our implementation the coefficient is set to zero. The global model also contains hyperdiffusion terms, but the coefficient was also set to zero. Please see the revised model descriptions. As is stated in the first paragraph of section 3.1, we consider the choice of diffusion/viscosity terms a modeling choice, but we acknowledge the other approach of specifying them as part of the test case.

12. Page 9, Fig. 2: The value for the symbol $\varphi$ f is not provided. Add this information.

$\varphi_f$ is removed from the text of the revised version.

13. Page 9, Fig. 2: It is highly unusual and confusing to see and interpret the flipped Hovmoeller diagrams. Typically, Hovmoeller diagrams list the position along the x-axis and time along the y-axis. I recommend flipping the axes in Fig. 2 to make the interpretation of the Hovmoeller diagrams easier.

To conform to common practice we changed the longitude-time diagrams into time-longitude diagrams.

14. Supplemental material: Please add Fortran/Matlab wrapper codes that will enable the user to create/test the initial conditions. In addition, the codes should not expect to receive regular longitude and latitude arrays, but should be callable for any longitude and latitude position.

Thank you, but we have decided to leave the computation of the initial conditions to the developers that can do it better than us and in a way that suits their particular needs, e.g. unstructured meshes.

---

## Author Comment (AC4) · 8 Jan 2019

Please see our detailed response in our AC3

———————————————

---

## Referee Report (RR1)

Review of the revised GMDD manuscript 'The Matsuno baroclinic wave test case'
Authors: Ofer Shamir, Itamar Yavoby, and Nathan Paldor

**General comments:**
The manuscript introduces a test case for shallow water models that is built upon analytic
solutions of the linearized shallow water equations on the equatorial beta-plane, originally
published by Matsuno (1966). The research is very interesting and will add a very valuable
contribution to the literature.
The authors thoroughly revised the manuscript based on the first round of reviews. The revised
version has addressed many of my questions and concerns, but a few inconsistencies, a potential
initial data code bug, and omissions still remain in the manuscript. It needs to be revised further.
For example, a reader of this manuscript is still not able to initialize a shallow water model
without additional information from the initialization codes and there is a mismatch between the
initial data codes and Eq. (6a) in the manuscript. These issues are all listed in the specific
comments 1-7 below. Even after further corrections of the manuscript, it will still be paramount
to also publish the codes. I strongly argue against dropping the codes as the authors suggested.

Unfortunately, the manuscript creates a big new question mark. I had requested that the authors
also demonstrate that the test case can be used in global shallow water models on the sphere, not
just in tropical channel models (which are very rare). My expectation was that the results are
more or less identical to the tropical channel model, and wanted to see the confirmation. I am
now stunned to see completely different results for the global shallow water model, and the
discussion of the new global results is extremely sparse and insufficient. Why are the results so
different? The authors do not even comment on the fact that e.g. all colors are flipped when
comparing the Rossby wave in the tropical channel model (Fig. 1) to the Rossby wave of the
global shallow water model (Fig. 4). I do not see a reason why even the 'initial states'
(comparing hour 0 in Fig. 1 to hour 4 in Fig. 4) show opposite colors. The 4-hour difference in
the time snapshots, which I find unnecessary and very irritating, cannot be the reason due to the
long wave period of 18.5 days. This looks like a sign error in the initialization to me. I therefore
suspect that one of the model setups (either the channel or the global) were initialized incorrectly
and need to be reassessed. More details are provided in the specific comments 12-14 below.

**Specific comments:**

1) Page 3, Eq. (1): Add the definition of the imaginary unit $i$. In addition, the manuscript (Eq.
(1)) fails to state that only the real parts of Eq. (1) and also Eq. (3) are used for the initialization
of u, v, $\Phi$, $\omega_{n,k,j}$ as coded in the initialization codes.

2) Page 3, line 19: The manuscript omits the definitions of the physical constants $\Omega$, $a$ and $g$.
Please add these to promote completeness and reproducibility, since there are many possibilities
to set these constants. I assume that it is paramount that the tested shallow water model also
needs to use the same constants. Make the user aware of this.

3) Page 4, Eq. (6) and line 20: The expressions for the symbols $H_n$ and $\hat{H}_n$ are missing, and the
connection between $H_n$ (in Eq. (6)) and the normalized $\hat{H}_n$ (line 20) is unclear. Do you imply $H_n$
$=\hat{H}_n$ as suggested by the code? In case they are the same why are two different symbols used
(maybe typo)? The manuscript points to the Press et al. (2007) 'Numerical Recipes' book for the

explanation of the Hermite three-term recurrence relation for $H_n$. I think this is a major barrier for the adoption of this test case by others, especially if the publication of the initialization codes is dropped. Taking a look at the initialization codes, the exact definition and normalizations for $H_n$ used here are short enough so that they should be provided in an Appendix to this manuscript. Please add this information for completeness.

4) Page 4, Eq. (6a): There is a mismatch between the definition of $\hat{v}_n$ (formerly $\psi_n$ (Eq. (4)) in the first version of the manuscript) and the Fortran/Matlab/Python initialization codes. In the original Fortran/Matlab/Python initialization codes (e.g. line 158 in the Fortran program) $\psi_n$ is initialized as

$$(\hat{v}_n =)\ \psi_n = AH_n \exp\left[-\frac{1}{2}\varepsilon^{1/2}\left(\frac{y}{a}\right)^2\right]$$

but the manuscript defines this quantity as (see the definition of $\hat{v}_n$ in Eq. (6a)):

$$\hat{v}_n = \psi_n = AH_n\left[\varepsilon^{1/4}\left(\frac{y}{a}\right)\right]\exp\left[-\frac{1}{2}\varepsilon^{1/2}\left(\frac{y}{a}\right)^2\right]$$

The factor $\varepsilon^{1/4}\left(\frac{y}{a}\right)$ is missing in the codes. Is this a typo in the manuscript or an error in the initialization codes? If it is an error in the initialization codes then all simulations and analyses need to be repeated.

5) Page 6, line 8: There is a wrong definition of the planar wavenumber k (with units 1/m). The parameter k is confused with its dimensionless spherical counterpart $k_s$. The definition k=5 needs to read $k_s$ =5 and, for completeness, I recommend also providing the definition of k= $k_s$/a (for the equatorial plane) again.

6) Page 6, line 13: When providing the codes, please make sure that the parameters in the codes match the manuscript, e.g. correct the amplitude to amp=$10^{-5}$ m/s in the shallow water model (line 117) to match the information for the amplitude A on page 6, line 13.

7) Page 8, line 7: The manuscripts fails to describe the boundary conditions (in the y-direction) for u and $\Phi$. Please add this information for the channel model.

8) Page 8, lines 31-33, Fig. 2 (g): What does 'appears to be' mean? You seem to suggest that a tiny phase speed error leads to an exact $\pi/4$ (or any integer multiple of $\pi/4$ ?) shift of the v latitude-time diagram (panel g) after 99 wave periods. I find this speculation questionable, and even if true it would not be a tiny error. If there were a phase error like this, why would this not affect all other fields as well? The authors lost me here. Could there be a plotting problem with panels 2g and also 2j (e.g. wrong output file)?

9) Page 10: add a reference for the GFDL spectral transform shallow water model

10) Page 11, line 7: remove double 'the'

11) Page 11, line 8: remove 'at'

12) Page 11: Almost all users of the test case will use a global shallow water model, so the addition of the new section 4.2 is very much appreciated. However, the current discussion of the global shallow water results is inadequate and insufficient. In my view, the discussion of the global results is way more important than the discussion of the tropical channel results, but this section seems to be added in a rushed fashion. As mentioned above, I am now stunned to see completely different results for the global shallow water model. Why are the results so different in comparison to the channel model?

The authors do not even comment on the fact that e.g. all colors are flipped when comparing the Rossby wave in the tropical channel model (Fig. 1) to the Rossby wave of the global shallow water model (Fig. 4). The same is true for the EIG wave (Figs. 2 and 5). I suspect this is due to an initialization (sign) error in either the channel or the global model. This needs to be investigated, likely reassessed and explained.

13) Figs. 4 and 5 (in comparison to Figs. 1 and 2): I do not see a reason why even the 'initial states' (e.g. comparing hour 0 in Fig. 1 to hour 4 in Fig. 4) show opposite colors. The 4-hour difference in the time snapshots, which I find unnecessary and very irritating, cannot be the reason due to the long wave period of 18.5 days of the Rossby wave. As mentioned in point 12) this looks like a sign error in the initialization to me. Change Figs. 4 and 5 and show time step 0 instead of the simulation after 4 hours. In addition, replot Fig. 4 and label the 18W and 18E points (as in all other figures) instead of the 15W and 15E points.

14) Figs. 4 and 5: Figs. 1 and 2 are normalized with the initial states at t=0. Did you normalize Figs. 4 and 5 with the state after 4 hours or with the state at t=0? If the state after 4 hours was used, correct the normalization (use t=0) to make the normalization procedures identical.

15) Page 14, line 14: correct 'the fact that all …'

16) Page 15: The new spatial convergence study is helpful, but the way it was conducted is questionable. In order to test spatial convergence, one typically selects the smallest time step (to minimize time step errors) and keeps this time step constant. Instead, the authors used the longest time step, kept it constant for decreasing grid spacings and of course observed numerical instabilities at some point. The time step errors could potentially be very large for the shortest grid spacings, and these errors become part of the spatial convergence assessment. I suggest repeating the convergence study and using a very small time step for all simulations to avoid this problem.

---

## Author Response (AR2)

**Authors response to reviewer's and editor's comments on GMD 2018-260 "The Matsuno baroclinic wave test case" by Shamir, Yacoby, Ziskin and Paldor**

immediate

**Correspondence:** Nathan Paldor (nathan.paldor@huji.ac.il)

**General response**

We appreciate the reviewer's meticulous evaluation of both the text and the code. We provide a detailed response to each of his/her comments below. Save for a couple of graphical errors that resulted in the opposite colors between Figs. 1,2 and Fig. 4,5 and the revision of the convergence test, all other comments were found to be only minor comments, which were fully implemented in the revised version. In particular, the potential bug flagged by the reviewer resulted from a misunderstanding and required only clarification in the text.

In order to address the editor's technical comments we have recruited the help of an expert on this issue, who was responsible for packaging the code, verifying (i.e. unit testing), interfacing, licensing and archiving. Consequently, this expert was added as a co-author in acknowledgement of his significant contribution. Please see our detailed response below, following our detailed response to the reviewer's comments.

**Response to the specific comments made by the reviewer**

1) Page 3, Eq. (1): Add the definition of the imaginary unit i.

Added - see paragraph following Eq. (1) of the revised manuscript

In addition, the manuscript (Eq. (1)) fails to state that only the real parts of Eq. (1) and also Eq. (3) are used for the initialization of $u$, $v$, $\Phi$, $\omega_{n,k,j}$ as coded in the initialization codes.

Added - see Eqs. (1) and (3) of the revised manuscript

2) Page 3, line 19: The manuscript omits the definitions of the physical constants $\omega$, $a$ and $g$. Please add these to promote completeness and reproducibility, since there are many possibilities to set these constants. I assume that it is paramount that the tested shallow water model also needs to use the same constants. Make the user aware of this.

We assume that the reviewer means the actual numeric values (the physical definition were already given). These values were added as requested - see the definitions following Eq. (2) in the revised manuscript. (We note that since the test is analytic

and not an inter-comparison one if different developers use slightly different values they can still compare their simulations to the analytic results provided they use those value in the analytic expressions as well and $H$ is chose so as to ensure that $gH$ is small.)

3) Page 4, Eq. (6) and line 20: The expressions for the symbols $H_n$ and $\hat{H}_n$ are missing, and the connection between $H_n$ (in Eq. (6)) and the normalized $\hat{H}_n$ (line 20) is unclear. Do you imply $H_n = \hat{H}_n$ as suggested by the code?

The code does not imply that $\hat{H}_n = H_n$. What determines the normalization of the calculated polynomials is the specific three-term recurrence relation (including its initialization). The three-term recurrence relation used in the code corresponds to the normalized Hermite polynomials (normalized in the sense that they are orthonormal). The non-normalized one satisfy a slightly different relation.

In case they are the same why are two different symbols used (maybe typo)?

Yes it is a typo, sorry. there is no need for the non-normalized polynomials $H_n$ at all, only the normalized ones $\hat{H}_n$ are used. Corrected - see Eq. (6a) of the revised manuscript.

The manuscript points to the Press et al. (2007) 'Numerical Recipes' book for the explanation of the Hermite three-term recurrence relation for $H_n$. I think this is a major barrier for the adoption of this test case by others, especially if the publication of the initialization codes is dropped. Taking a look at the initialization codes, the exact definition and normalizations for $H_n$ used here are short enough so that they should be provided in an Appendix to this manuscript. Please add this information for completeness.

Actually, the three-term recurrence relation is short enough and is included in the main text (and too short for an appendix). Please see the new paragraph following Eq. (6) and the new Eq. (7) of the revised manuscript.

4) Page 4, Eq. (6a): There is a mismatch between the definition of $\hat{v}_n$ (formerly $\psi_n$ (Eq. (4)) in the first version of the manuscript) and the Fortran/Matlab/Python initialization codes. In the original Fortran/Matlab/Python initialization codes (e.g. line 158 in the Fortran program) $\psi_n$ is initialized as

$$(\hat{v}_n =)\psi_n = AH_n \exp\left[-\frac{1}{2}\epsilon^{1/2}\left(\frac{y}{a}\right)^2\right] \tag{1}$$

but the manuscript defines this quantity as (see the definition of $\hat{v}_n$ in Eq. (6a)):

$$\hat{v}_n = \psi_n = AH_n\left[\epsilon^{1/4}\left(\frac{y}{a}\right)^2\right]\exp\left[-\frac{1}{2}\epsilon^{1/2}\left(\frac{y}{a}\right)^2\right] \tag{2}$$

The factor $\epsilon^{1/4}\left(\frac{y}{a}\right)$ is missing in the codes. Is this a typo in the manuscript or an error in the initialization codes? If it is an error in the initialization codes then all simulations and analyses need to be repeated.

The quantity in square brackets in Eq. (6a) is not a multiplicative factor. It is the argument of the Hermite function $H_n$ ($\hat{H}_n$ in the revised version), which was denoted with square brackets instead of the more conventional parentheses due to the use of the latter to encapsulate (y/a). In order to simplify the expression we removed the parentheses from (y/a) and use them for the

function's argument as is more common. We also added a comment in the text that explains it - see note (iii) after Eq. (7) in the revised manuscript. In other word, the independent variable in Eq. (6a) is $(\epsilon^{1/4}y/a)$, and not just $y$. This is also the way it is coded. The function get_hermite_polynomial(x,n) (e.g. lines 116-134 of the Fortran code) is defined for arbitrary independent variable, but when it is called (e.g. in line 158 of the Fortran code) it is called for the independent variable $(\epsilon^{1/4}y/a)$ (note the definition of $y$ in line 153).

5) Page 6, line 8: There is a wrong definition of the planar wavenumber k (with units 1/m). The parameter k is confused with its dimensionless spherical counterpart ks. The definition k=5 needs to read ks =5 and, for completeness, I recommend also providing the definition of k= ks/a (for the equatorial plane) again.

Done - third paragraph in Section 3.1 of the revised manuscript

6) Page 6, line 13: When providing the codes, please make sure that the parameters in the codes match the manuscript, e.g. correct the amplitude to amp=10-5 m/s in the shallow water model (line 117) to match the information for the amplitude A on page 6, line 13.

Done. The default amplitude is now set to e-5 m/s.

7) Page 8, line 7: The manuscripts fails to describe the boundary conditions (in the y-direction) for $u$ and $\Phi$. Please add this information for the channel model.

Matsuno's solutions are derived from a second order equation for $v(y)$ and therefore the **only** boundary conditions required to close the problem are $v(y) = 0$ at the channel walls. The corresponding values of $u$ and $\Phi$ at the channel walls are determined directly from the differential equations themselves.

8) Page 8, lines 31-33, Fig. 2 (g): What does 'appears to be' mean?

We meant that it appears to be $\pi/4$, but can be any integer multiple of $\pi/4$. In any case we have rephrased the description of Fig. 2 - please see revised discussion of Fig 2. starting on p.9

You seem to suggest that a tiny phase speed error leads to an exact $\pi/4$ (or any integer multiple of $\pi/4$ ?) shift of the $v$ latitude-time diagram (panel g) after 99 wave periods. I find this speculation questionable, and even if true it would not be a tiny error.

We have rephrased the description of Fig. 2 - please see revised discussion of Fig 2. starting on p.9

If there were a phase error like this, why would this not affect all other fields as well? The authors lost me here. Could there be a plotting problem with panels 2g and also 2j (e.g. wrong output file)?

It does affect all other fields. The phase shift between the simulated patterns and the expected ones is the same in all panels (f)-(j) of Figure 2. Note that, while all Hovmoller diagrams are generated at the same longitudinal intersect $\lambda = -18°$ degrees, this intersect corresponds to a different phase in each of the different initial fields shown in the top row of the figure, as indicated by the vertical white dashed lines. In other words the initial fields themselves have different phases (which is a general property of the SWEs). We originally wrote this discussion for panel (g) thinking that it will simplify the discussion, since $v$ is independent of the wave type (i.e. it is identical in Rossby and IG modes). Since it ended up being more confusing, we do not single out panel (g) anymore in the revised version - please see revised discussion of Fig 2. starting on p.9

9) Page 10: add a reference for the GFDL spectral transform shallow water model

Unfortunately, this model has no proper reference. We provide a link to its online description on GFDL's website -see first paragraph of Section 4.2.

10) Page 11, line 7: remove double 'the'

Done - thank you

11) Page 11, line 8: remove 'at'

Done - thank you

12) Page 11: Almost all users of the test case will use a global shallow water model, so the addition of the new section 4.2 is very much appreciated. However, the current discussion of the global shallow water results is inadequate and insufficient. In my view, the discussion of the global results is way more important than the discussion of the tropical channel results, but this section seems to be added in a rushed fashion. As mentioned above, I am now stunned to see completely different results for the global shallow water model. Why are the results so different in comparison to the channel model? The authors do not even comment on the fact that e.g. all colors are flipped when comparing the Rossby wave in the tropical channel model (Fig. 1) to the Rossby wave of the global shallow water model (Fig. 4). The same is true for the EIG wave (Figs. 2 and 5). I suspect this is due to an initialization (sign) error in either the channel or the global model. This needs to be investigated, likely reassessed and explained.

The reviewer is right. We expect only little difference between the global model and the equatorial-channel one, provided the channel is sufficiently wide so that the solutions are not affected by the imposed boundary conditions at the walls which are not included in Matsuno's theory. Unfortunately, there were a couple of plotting error that resulted in the opposite signs: 1) The definition of the longitude in the two models is different. The longitude in the equatorial-channel model is defined for $\lambda_1 \in [-180, 180]$, whereas the longitude angle in the global model is defined for $\lambda_2 \in [0, 360]$. Note that $\cos(5\lambda_1) = \cos(5(\lambda_2 -$

$\pi)) = -\cos(5\lambda_2)$. 2) The direction of the latitudes in Fig. 1 and 2 was accidentally (bug-like) flipped. For the chosen parameters initial $v$ field should increase with latitude at $\lambda = 0$.

Despite the unfortunate plotting errors, an opposite sign is hardly a mistake in our view since for linear waves the amplitude is arbitrary so the simulations work just as well with an opposite sign. The only restriction on the amplitude is that it's sufficiently small for the non-linear terms to be negligible, but not too small so as to avoid round-off errors - both of which stem from practical considerations.

Having fixed the graphical errors, the solutions of the global model are indeed similar to those of the equatorial channel. At the risk of being repetitive and in order to respond positively to the reviewer's concerns we repeat some of the explanation from the subsection on the equatorial channel in the spherical model subsection so the two sections can be read independently - please see the revised section 4.2.

13) Figs. 4 and 5 (in comparison to Figs. 1 and 2): I do not see a reason why even the 'initial states' (e.g. comparing hour 0 in Fig. 1 to hour 4 in Fig. 4) show opposite colors. The 4-hour difference in the time snapshots, which I find unnecessary and very irritating, cannot be the reason due to the long wave period of 18.5 days of the Rossby wave. As mentioned in point 12) this looks like a sign error in the initialization to me.

Unfortunately, there were a couple of plotting error that resulted in the opposite signs: 1) The definition of the longitude in the two models is different. The longitude in the equatorial-channel model is defined for $\lambda_1 \in [-180, 180]$, whereas the longitude angle in the global model is defined for $\lambda_2 \in [0, 360]$. Note that $\cos(5\lambda_1) = \cos(5(\lambda_2 - \pi)) = -\cos(5\lambda_2)$. 2) The direction of the latitudes in Fig. 1 and 2 was accidentally (bug-like) flipped. For the chosen parameters initial $v$ field should increase with latitude at $\lambda = 0$.

Change Figs. 4 and 5 and show time step 0 instead of the simulation after 4 hours. In addition, replot Fig. 4 and label the 18W and 18E points (as in all other figures) instead of the 15W and 15E points.

Done - see revised version of the figures

14) Figs. 4 and 5: Figs. 1 and 2 are normalized with the initial states at t=0. Did you normalize Figs. 4 and 5 with the state after 4 hours or with the state at t=0? If the state after 4 hours was used, correct the normalization (use t=0) to make the normalization procedures identical.

Done - see revised version of the figures. This was also relevant to Figs. 7 and 8. Note that the differences in the divergence field in Fig. 7 compared to the previous manuscript version is the result of this change.

15) Page 14, line 14: correct 'the fact that all ...'

Fixed - thank you

16) Page 15: The new spatial convergence study is helpful, but the way it was conducted is questionable. In order to test spatial convergence, one typically selects the smallest time step (to minimize time step errors) and keeps this time step constant. Instead, the authors used the longest time step, kept it constant for decreasing grid spacings and of course observed numerical instabilities at some point. The time step errors could potentially be very large for the shortest grid spacings, and these errors become part of the spatial convergence assessment. I suggest repeating the convergence study and using a very small time step for all simulations to avoid this problem.

What do you mean the longest time step? It is not true that we took a time step for which the greatest grid spacing (lowest resolution) converges, held it fixed and decreased the grid spacing (increased the resolution). We did the opposite, we took the time step for which the results for the finest grid spacing in the previous sections converged and increased the grid spacing (lowered the resolution). Note that all the results of the previous sections, including both the finite difference model and the spectral one, were generated using the same time step of $\Delta t = 600$ sec used for the convergence analysis in Fig. 10 and using the finest grid spacing in this figure $\Delta = 0.5°$. Also how small is very small? In an attempt to address this comment we have repeated the convergence test using a time step of 100 sec (six times smaller) and replaced the results in the Figure, with the new ones. The results remained visually unchanged, but we were able to increase the resolution by another $0.25°$ and add another data point to the figure. We hope the chosen time step is small enough. We also added a comment on the issue - see revised Section 4.4.

**Detailed response to the technical issues made by the editor**

**Editor P1  Deployability:**

Individual source files in a paper supplement are not particularly easy for language users to employ. This is unfortunate because Python and Matlab (though not really Fortran) provide straightforward mechanisms for packaging files to make them easily installable and usable. In the case of Python, if the package files are then hosted on an online revision control system such as GitHub then users will be able to directly pip install them with a single command. Instructions on how to create a Python package are at https://packaging.python.org/tutorials/packaging-projects/ while Matlab toolboxes are at: http://www.mathworks.com/help/matlab/matlab_prog/create-and-share-custom-matlab-toolboxes.html.

**Reply**:  We agree with the editor on this important point. Thus, the Python code was packaged using PyPi service (https://pypi.org/) with the name `pymaws` (PYthon Matsuno Analytical Wave Solutions) and is now fully deployable using the command `pip install pymaws`.

**Editor P2  Verification and unit testing:**

There is currently no testing of the code provided. There is little more frustrating when using a test case than to eventually discover that the bug was actually in the test case. Potential users will therefore want to see that there is a robust set of verification tests that demonstrate that the implementation is correct. In this case you have two routes which can both be used.

The first is to check the output of your routines for parameter values where the output is easy to verify. The second is to cross-verify your code by calling all the implementations and checking that their output is the same. For this purpose, it is useful to note that the Matlab code might be callable from Python using https://pypi.org/project/oct2py/. It is possible to directly call Fortran from Python using f2py, or you could use the Fortran standard C interoperability features to present a public C interface (which would also be a nice user feature!) and call that using `ctypes`. There is some documentation on how to do this in the https://docs.scipy.org/doc/numpy-1.15.4/user/c-info.python-as-glue.html. You should use one or more test frameworks (maybe one, maybe you need one for each language) to run the testing in a way that users will understand and which is easy for them to integrate into their own testing frameworks. For example you might use https://pytest.readthedocs.io/en/latest/. There are likewise suitable testing frameworks available for Matlab and Fortran (or, if you set up cross-verification, you could just use the Python framework for all languages).

**Reply**: The new code has been thoroughly tested with Python's native `unittest` module. After the installation, the users can run `python test_pymaws.py` in order to verify the code works. Furthermore, all the test procedures are fully documented (i.e., with docstrings), so the users can inspect the test code and integrate it into their own testing frameworks.

**Editor P3  Code interfaces and naming:**

As far as I can see, the public interface of the code consists primarily of the function `get_fields`, which seems reasonable given that the purpose of the code is to execute one particular test case. However there are a number of things about this interface which are unhelpful. First, the name `get_fields` is problematic. It's not going to be obvious to the user that that is the main function and even what it does. A more descriptive name would be useful, potentially without the `get_` prefix, which adds no information. In Python, the auxiliary routines which are not supposed to be visible outside the package should be prefixed with underscore to mark them as private. In both Matlab and Python, the documentation for functions should be placed in the docstring at the start of the function, not at the start of the file. This change will enable the self-documentation features of these languages to work. It should also be borne in mind that users may see the code without the paper, so it would be a good idea if the docstrings referred back to the paper. The code returns values on a regular latitude-longitude grid. This is not useful for many users. Even many structured grid models have curvilinear grids which are not axis-aligned, and this is before considering unstructured mesh models. The functions should instead evaluate the results at an arbitrary set of latitude-longitude points. The modest performance gain that may be available from the current structure is not worth the cost of having code that many users simply will not be able to use. It is quite conceivable that the user will want to change $g$, $\Omega$, $A$, or $H0$ in order to run the test case for a model using scaled parameters. Currently in Fortran these are private, and in all languages this change would involve changing the global state of the module, which is an unsafe programming practice. It would be better if these parameters were optional arguments to the main function(s), with default values being those given currently. The test case is only valid for certain values of input parameters. In addition to documenting these limitations, the code should check for illegal values and raise appropriate exceptions or return some error, depending on language.

**Reply**: The main function was renamed to `eval_field` for more functional clarity. Furthermore, each function now is fully documented (i.e., with docstrings) and all the private functions were renamed with an underscore preceding their name, as in Python convention. As suggested by the editor, the new code's main function ( `eval_field` ) now evaluates the results at an arbitrary latitude-longitude point. The users can then call this function on each of their grid points, regardless of its structure. Moreover, the new code supports different input planetary parameters using a global dictionary named `Earth` (default parameters) which can be easily modified per user input. Finally, the new code's functions support raising exceptions for non-appropriate/not-supported user input. For example, the current version of the code supports $n, k >= 1$, so using an input of $n = 0$ will result in an error message.

**Editor P4  License:**

There is currently no license provided with the code. This might mean that the code falls under the same Creative Commons attribution license as the paper, but since the code may be separated from the paper when used, it would be preferable to provide an explicit license. Many authors would use the MIT License in this context, which basically says that users can do anything they like so long as they continue to acknowledge the authorship of the code. See https://opensource.org/licenses/MIT. In any event, an explicit license should be chosen and included with the source files.

**Reply**: The new code is now licensed under the MIT open-source license (https://opensource.org/licenses/MIT).

**Editor P5  GitHub and archiving:**

The steps above will result in quite a lot more files in the implementation, along with a nontrivial directory structure. This could be attached in the supplement, but this has a number of disadvantages. First, there is no mechanism for future bug fixes. Next, direct access mechanisms such as Python's package installer pip will not see the supplement. Instead, the code could be posted on GitHub, which would fix the update and pip problems. GitHub is not a suitable archive location for the canonical version of the code in a manuscript, but GitHub and Zenodo together provide a straightforward way of archiving the repository with a DOI for inclusion in the manuscript. The instructions are at https://guides.github.com/activities/citable-code/. GitHub + Zenodo would be a better alternative to the supplement.

**Reply**: Following the editor's suggestion the new code is now posted at GitHub.com (https://github.com/ofershamir/matsuno) and its documentation( `README.md` ) includes:

- Installation

- Testing

- Importing

- Example

- Caveats

Furthermore, the documentation demonstrates how to alter the planetary parameters if the user wishes to do so. Finally, as suggested by the editor, the new code release is now archived with Zenodo and can be cited using the DOI:10.5281/zenodo.2605203 (https://zenodo.org/record/2605203).

[revised manuscript text omitted]

---

## Author Response (AR3)

**Authors response to editor's final comments on GMD 2018-260 "The Matsuno baroclinic wave test case" by Shamir, Yacoby, Ziskin and Paldor**

immediate

**Correspondence:** Nathan Paldor (nathan.paldor@huji.ac.il)

**General response**

In response to the latest round of editor's comment we've checked the "My final revised paper has no supplement" option in the Supplement section. In the text itself (acknowledgement section): We added an acknowledgement of the insightful comments we received from the two reviewers.